# What Prevents the Adoption of Regenerative Agriculture and What Can We Do about It? Lessons and Narratives from a Participatory Modelling Exercise in Australia

Daniel C. Kenny [1,2] and Juan Castilla-Rho [3,4,*]

1. School of Information, Systems and Modelling, Faculty of Engineering and Information Technology, University of Technology Sydney (UTS), Sydney, NSW 2007, Australia
2. PERSWADE Research Center, Sydney, NSW 2007, Australia
3. Faculty of Business, Government & Law, University of Canberra (UC), Canberra, ACT 2617, Australia
4. Center for Change Governance (CCG), Institute for Governance and Policy Analysis (IGPA), Canberra, ACT 2617, Australia
* Correspondence: juan.castillarho@canberra.edu.au

**Abstract:** Regenerative agriculture (RegenAg) can help landholders attune their agricultural practices to the natural design of the earth's cycles and support systems. The adoption of RegenAg, however, hinges not only on a good understanding of biophysical processes but perhaps more importantly on deep-seated values and beliefs which can become an obstacle for triggering widespread transitions towards synergistic relationships with the land. We designed and facilitated a Participatory Modelling exercise with RegenAg stakeholders in Australia—the aim was to provide a blueprint of how challenges and opportunities could be collaboratively explored in alignment with landholders' personal views and perspectives. Fuzzy Cognitive Maps (FCM) were used to unpack and formalise landholder perspectives into a semi-quantitative shared 'mental model' of the barriers and enablers for adoption of RegenAg practices and to subsequently identify actions that might close the gap between the two. Five dominant narratives which encode the key drivers and pain points in the system were identified and extracted from the FCM as a way to promote the internalisation of outcomes and lessons from the engagement. The Participatory Modelling exercise revealed some of the key drivers of RegenAg in Australia, highlighting the complex forces at work and the need for coordinated actions at the institutional, social, and individual levels, across long timescales (decades). Such actions are necessary for RegenAg to play a greater role in local and regional economies and to embed balancing relationships within systems currently reliant on conventional agriculture with few internal incentives to change. Our methods and findings are relevant not only for those seeking to promote the adoption of RegenAg in Australia but also for governments and agriculturalists seeking to take a behaviorally attuned stance to engage with landholders on issues of sustainable and resilient agriculture. More broadly, the participatory process reported here demonstrates the use of bespoke virtual elicitation methods that were designed to collaborate with stakeholders under COVID-19 lockdown restrictions.

**Keywords:** regenerative agriculture; participatory modelling; fuzzy cognitive mapping; socio-ecological systems; stakeholder engagement

## 1. Introduction

In the middle of 2019, the Australian landscape began to burn. By the time the fires ran their course, over 240 days later, more than 30 people had died, 3500 homes had been destroyed, 306 millions tons of carbon dioxide had been released, and costs were approaching 100 billion [1–3]. The entire country was and, in many places, still is reeling from the devastation. With links to climate change increasing and suggesting the possibility

of a repeat in the future [4], serious questions confront both policymakers and Australian citizens about how this issue can be dealt with so as not to face this level of devastation ever again. This issue is particularly pertinent for farmers, a group severely affected by the fires, and the drought leading up to it [5–7]. Unfortunately for agriculture, as a system where social desires do not necessarily align with vested interests, current policy regimes, environmental trends, or market pressures, there are no simple solutions.

In the last few decades, Australian farmers have seen enormous changes in their farm systems, but also in the social, economic, and political systems that govern the land across the country [8]. Agriculture is inherently exposed to "multiple, simultaneous and interconnected ecological, economic and social pressures" [9]. The impacts of these pressures are typically seen over long time periods [10,11], as the lands on which they farm tend to be governed by 'slow variables', i.e., variables that are crucial to the health of the ecosystem but whose trends can only be understood in timeframes of decades or longer, despite short-term variations. These variables include climate patterns, (including rainfall), ground coverage of perennial species, local environmental and scientific knowledge, and others [8,11]. For example, fires are a random and natural occurrence, but the severity and frequency of fires in Australia can be determined by trends (such as fire prevention efforts near populated areas, or a lack of backburning) decades in the making.

Farm ecosystems are shaped by these slow variables, which have their own natural trends, but farms are also under increasing and more immediate pressure from human interventions. The complexity of all these interactions makes farms a difficult system to manage; there is simultaneously a resilience and a fragility to these tightly linked ecological, economic and social systems [12,13]. For example, fires can carry serious consequences by altering groundcover, changing the physical properties of the soil (including hydrologic properties), altering the composition of soil microbial communities, changing and altering the land cycles of carbon and nitrogen fixation, and ultimately, reducing the number of plants holding the soil in place [14]. This makes erosion more likely and the land more susceptible to flooding. These chains of impact carry implications for people's livelihoods by ultimately affecting farm productivity [10]. As such, we need an understanding of both the thresholds and non-linear trends in these complex, socio-environmental systems, and crucially, the role of the individuals within it, as it is their preferences and decisions that shape these farm ecosystems [15–17].

Policymakers have designed and implemented a wide range of agricultural policies attempting to address issues of erosion, water pollution, climate change, and other related issues. These policies, however, often fail to account for the perceptions and beliefs of individual farmers (who have the final word on whether and how the policies are implemented), and therefore, they often fail to make a lasting impact [18]. When farmer perceptions are considered, it is often with a simplistic profit-driven motive, which has time and time again been shown to be misguided and overly simplistic [19–24]. Lessons from practice and scholarship show that farmers consider multiple factors in their decision making beyond money, including environmental stewardship, family legacy, and community [9].

As scientists and policymakers seeking to support enduring transitions toward sustainable agriculture, we need to develop a holistic understanding of ecological, environmental, and social factors and how they shape the preferences and motivations of farmers [17,18]. This is necessary, because ultimately, farmers are the agents undertaking action, and therefore, their buy-in or inaction directly determines the success of any sustainability program or policy on the ground [9]. If we understand this, then we can better design incentives, regulations, and institutional reforms, as well as choosing times when it is better not to be involved at all [9].

In this paper, we argue that by focusing on the perceptions and motivations of farmers through increasing our understanding of the stories they tell and the systematic connections within those stories, there is an increased likelihood of creating a more enduring form of bottom–up change in a new social norm rather than a top–down policy or incentive program that is subject to change with each new election cycle [25]. This does not mean that

existing policies or incentives should be abandoned, but if they come from a ground–up understanding of farmer decision making and motivations, they may be more likely to be adopted by farmers in the first place and to endure beyond the limits of a given political cycle. To test and demonstrate the potential of adopting this approach, we conducted a participatory Fuzzy Cognitive Mapping exercise which unpacks stakeholder perspectives into a formal representation or 'mental model' of the barriers and enablers for adoption of RegenAg practices. To promote a better understanding and internalisation of the outcomes of the engagement, we extracted the dominant narratives which encode the pain points and leverage opportunities in the system. We report on a suite of bespoke virtual delivery methods that were designed to conduct the stakeholder engagement under lockdown and social distancing restrictions due to the COVID-19 pandemic. Our methods and findings are relevant not only for those seeking to promote adoption of RegenAg in Australia, but more broadly for practitioners, for researchers, and for government officials seeking to take a behaviorally attuned PM stance to engage with their stakeholders in geographically dispersed and value-laden SES contexts.

The paper proceeds as follows. Section 2 outlines the case study. Section 3 describes the participatory modelling process that was undertaken. Section 4 presents a macro, meso, and micro-scale analysis of the process and the co-construction of narratives. Section 5 discusses the specific implications for RegenAg in Australia and broader implications for Participatory Modelling practice, limitations, and future research. Section 6 Concludes.

## 2. Case Study: The Mulloon Institute and Regenerative Agriculture

The Mulloon Creek Catchment is located in the Southern Highlands of New South Wales and is part of the traditional country of the Yuin people, covering an area of 23,000 hectares and comprising more than fifty kilometres of creeks and tributaries, and four floodplains (Figure 1). Mulloon Creek Catchment feeds into the Shoalhaven River, which forms a vital source for Sydney's drinking water. The landscape of the catchment has historically been associated with pasture production for both sheep and cattle [26].

The Mulloon Rehydration Initiative, run by the Mulloon Institute, is a catchment-scale land management project. The project is a collaboration of 20 private landholders, comprising both production and amenity landholders. It aims to rebuild the natural landscape function of the entire Mulloon catchment to boost its resilience to climatic extremes through more reliable stream flows, improved ecosystem functioning and enhanced agricultural productivity. It does so through an approach called 'Natural Sequence Farming' (NSF), which falls under the larger umbrella of RegenAg approaches. In contrast to conventional agricultural techniques which may focus on a mechanistic and reductionist approach to maximal production, RegenAg methods instead focus on aligning with landscape function, regenerating biodiversity, and partnering with animals, microbes, and pollinators for a more holistic and resilient approach [27]. While there are many techniques, practices, traditions, and definitions within RegenAg, for this paper, we define RegenAg broadly as an "alternative form of food and fiber production, concern[ed] with enhancing and restoring resilient systems supported by functional ecosystem processes and healthy, organic soils capable of producing a full suite of ecosystem services, among them soil carbon sequestration and improved soil water retention" [21].

Within RegenAg, NSF focuses on the vegetation, the daily water cycle, and the hydrology of the area, as these are the three critical areas controlling the landscape. Among other things, practitioners of NSF use vegetation, namely deep-rooted perennials to draw and store water deep within the land [28]. Mulloon Creek has trialled NSF since 2005. There were significant challenges in garnering support, both from the local community and government agencies, but eventually, the Mulloon Rehydration Initiative was formed in 2016. The Project and TMI employ a number of RegenAg practices, including NSF, while also drawing on indigenous expertise and scientific research to measure the biophysical, economic and social impacts of its practices, including the implementation of formal scientific instrumentation and monitoring [29]. The Project has received enough attention that

the United Nations Sustainable Development Solutions Network chose Mulloon as one of five global demonstration projects for sustainable and productive farming [29], and they were recently awarded a $3.8 million dollar grant by the federal government to demonstrate the effectiveness of rehydration activities and train and educate land managers in holistic management, NSF, and regenerative agricultural practices [30].

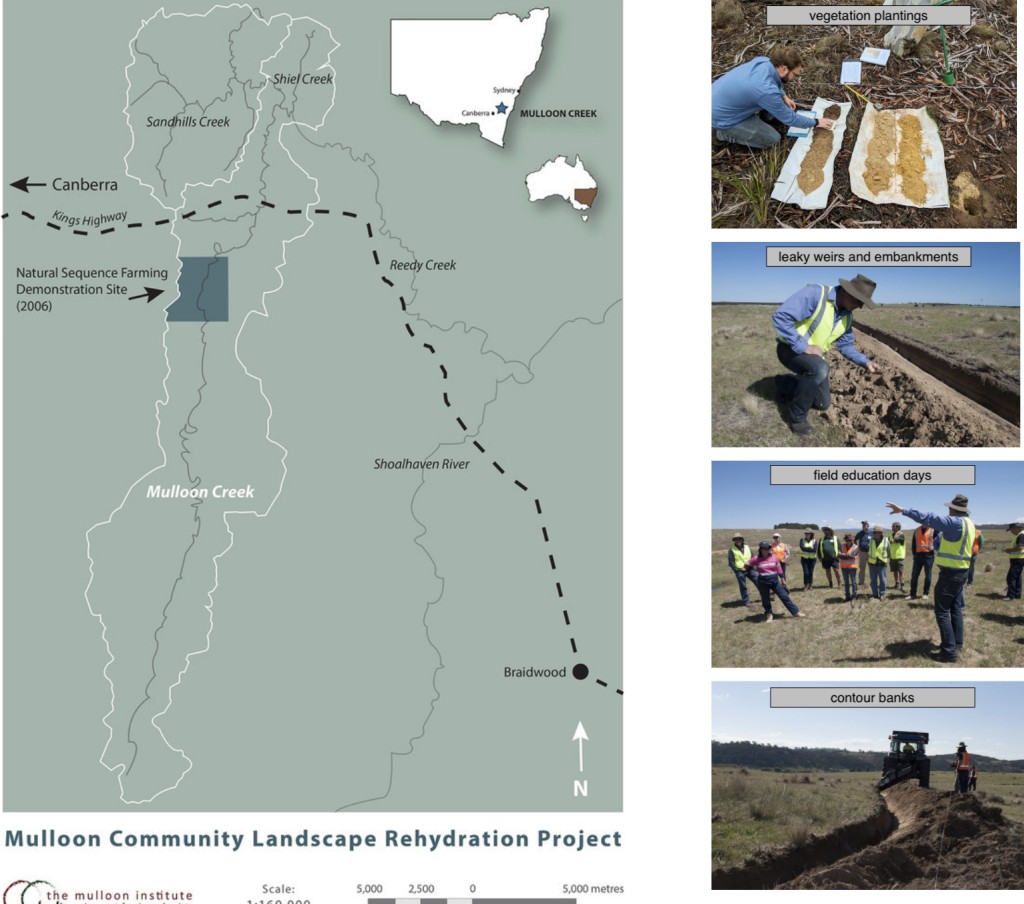

**Figure 1.** The Mulloon Creek catchment and land stewardship activities conducted by the Mulloon Institute (TMI), located near Braidwood in New South Wales. TMI focus on landscape rehydration activities, using Natural Sequence Farming. Activities include building leaky weirs, contour banks, embankments, and vegetation plantings, among others and have been linked to raising the water table, improving biodiversity, decreasing erosion, and building drought resilience.

TMI is a leader in this space, and outreach and education is a huge part of their portfolio. However, TMI and other RegenAg groups and leaders still face an uphill battle when it comes to adoption. Books such as Charlie Massy's *The Call of the Reed Warbler* point out the urgent need to make this transition, and it details some of the barriers to doing so based on nearly 80 interviews with farmers around Australia who had made the transition to RegenAg [31]. However, as the adoption of RegenAg has increased, and the evidence base continues to grow, the question has started to move from "Is RegenAg good?" to "How do we convince more people to do it?" The latter is a horse of an entirely different colour, but it holds serious implications for the future of farming in Australia.

## 3. Materials and Methods

### 3.1. Participatory Modelling and Fuzzy Cognitive Maps

Participatory Modelling (PM) is an umbrella term for tools and methods to place people at the centre of a scientific modelling process. In PM, stakeholders (i.e., members of the system of interest) build the model with researchers [32–34]. Regardless of the particular method chosen (system dynamics, agent-based modelling, causal loop diagrams, etc), PM seeks to bring researchers and stakeholders together on (at least) close to equal footing, in pursuit of 'shared understanding' of a problem, the system(s) within and around that problem, and the key components and relationships that combine to build those systems [32,35]. For this paper, we define PM as: "a purposeful learning process for action that engages the implicit and explicit knowledge of stakeholders to create formalised and shared representations of reality" [36]. In effective PM exercises, participants are empowered to ask questions, find answers, and make key decisions in the model-building process, in consultation and collaboration with researchers [37,38]. When dealing with SES, this style of participation can become a vehicle to elicit both tacit knowledge (qualitative and hard to verbalise) and scientific expertise. When combined, these two types of knowledge can substantially improve and inform both the model and process used to develop that model [39].

Within PM, Fuzzy Cognitive Mapping (FCM) is a tool commonly used [40–47]. FCM arose from the work of Axelrod, who introduced the idea of having stakeholders create 'cognitive maps' of a system for the purposes of social science research, which was later built upon by Kosko [48] who added fuzzy logic as a quantitative means to handle "vague and qualitative knowledge" [40,45,49]. FCM is a semi-quantitative knowledge elicitation technique used to represent the 'mental model' of an individual or a group—this takes the form of a qualitative 'map' of how someone believes a given system functions by identifying the variables or concepts of the system and relationships between them [50]. The quantitative element of the knowledge elicitation process comes from the way relationships are encoded, which can be either positive (>0), indicating an increase in A increases B, or negative (0<), indicating an increase in A results in a decrease in B [45]. The weights given to those relationships indicate the 'strength' of the causal relationships [51]. Overall, the focus of the FCM exercise is on identifying key feedbacks of the system to illustrate what variables are present in the system and how they affect each other [49]. The FCM can be created by an expert or a stakeholder, as an individual or as a group, and once the maps are drawn, the structure can be quantitatively analysed using graph theory [46].

FCM can have different uses, different purposes, and be appropriate for a few different situations. It can be used as a model of a particular system, and the end goal is to gain insight into the system of study; it can be used to drive communication and reduce conflict between stakeholders or different interest groups (policymakers, scientists, stakeholders, etc.); or it can serve as an initial participatory step in the building of a formal simulation model [52,53]. FCMs can be exploratory, as an attempt to understand the reasoning behind human behavior and actions or the wants and needs of stakeholders [40,54,55]. FCMs can also be used to create a portrayal of the system and to test how that system might act under different scenarios, which is particularly useful for testing different policy options and developing strategy [52]. FCM is generally an adequate engagement tool "(i) when dealing with complex problems; (ii) in situations where human behavior is important but hard to quantify; (iii) in situations where personal knowledge is available while scientific knowledge is incomplete; (iv) in situations where problems are wicked, involving many parties and with no easy solutions;" and (v) when the problem requires public involvement (possibly mandated by law) [52].

When it comes to SES as complex adaptive systems, FCMs can be useful and effective in guiding communication, comprehension, and problem solving without the use of complex mathematics [50,51]. However, they are not without their challenges and shortcomings, among them arguably, the main ones are the difficulties humans face in trying to share

their perspective on the system that is distorted by their own biases and values [52] and the limits of human knowledge [55]. These challenges can be addressed by combining the cognitive maps of individuals and by cross-checking with other sources of information and methodologies (interviews, surveys, etc.) to improve the accuracy of any 'map' or model [52]. FCM also cannot include aspects of time (including delayed feedbacks) [54], cannot handle Boolean expressions [56], struggles to provide insight behind the 'why' agents perceive the system as they have drawn it [57], and the nature of a causal relationships between variables can be positive or negative but not both [56].

Despite these challenges, FCMs offer advantages that make them well suited for tackling the complex socio-environmental problems Australian farmers face. FCMs are easy to teach and easy to use; they offer a systematic way to incorporate qualitative concepts into modeling a system and provide a clear representation of system feedbacks with a short turnaround [49,50,52]. FCM also works well with data that might be missing, is not well-defined, or might be uncertain [52]. The flexibility of the method allows for input from any number of stakeholders and experts, thus encouraging communication between and across diverse areas of knowledge [49], which can in turn stimulate a productive environment to test interventions and policy scenarios in the aims of seeking better management [50,54]. To summarise, FCM provides an efficient and useful methodology that can handle complex, uncertain systems that cross various fields of knowledge, which is precisely the type of problems Australian farmers face in their socio-environmental landscapes.

*3.2. Stakeholder Participation: Virtual Workshop in a Pandemic*

While we originally planned for our FCM workshops to be face-to-face, COVID-19 and NSW government regulations on social distancing in mid-2020 made this impossible. At the time, the restrictions limited the number of people gathering in one place, closed gyms, reduced the availability of public transit, and restricted seating in cafes and restaurants (NSW Government: Health, 2021). After some deliberation, the research team decided to pivot to an entirely virtual workshop. Due to the widespread availability of free team collaboration and video conferencing software, we were able to hold the workshop in an entirely digital format using a combination of Zoom and MURAL (a digital workspace and 'virtual whiteboard' for collaboration [58]). We used Zoom videoconferencing to hold the call with everyone and then used MURAL (www.mural.co, accessed on 10 August 2022) [58] for the FCM exercise. The benefit of this virtual approach was that it allowed us to engage with our stakeholders during a time of lockdowns and social distancing, but there were disadvantages as well, which were largely due to technical problems of unstable internet.

For our workshop, we invited fifteen participants, and while thirteen accepted, eleven ended up participating with two late withdrawals. Participants were a mix of researchers (6), farmers (6), and educator/trainers (4) of RegenAg, (*not exclusive categories*), who identified as male (8) and female (3), ranging in farming experience from none to over 30 years, and all based in New South Wales. TMI had 3 representatives, and 1 landholder from the Mulloon Creek Catchment also participated. To select the participants, we used purposeful sampling based on their affiliations with RegenAg, as we wanted a spectrum of practitioner, trainer, and researchers and educators of RegenAg in the workshop. This was in line with our qualitative approach and our aim to understand the underlying problems and barriers facing the adoption of RegenAg, as those advocates understood it (Bryman 2008; Bussing 2019). We personally contacted each participant and we obtained informed consent from all to participate, agreeing to protect their identities and keep their data confidential. This research was approved by the Human Research Ethics Committee at the University of Technology Sydney (ETH19-3712). For participating academics, permission was also secured from their respective universities to participate.

The aim of the workshop was to elicit mental models of the barriers facing the adoption of regenerative agriculture via facilitated stakeholder interaction. To do so, we designed a

virtual workshop consisting of four stages: (1) plenary, (2) elicitation, (3) modelling, and (4) debrief (Figure 2).

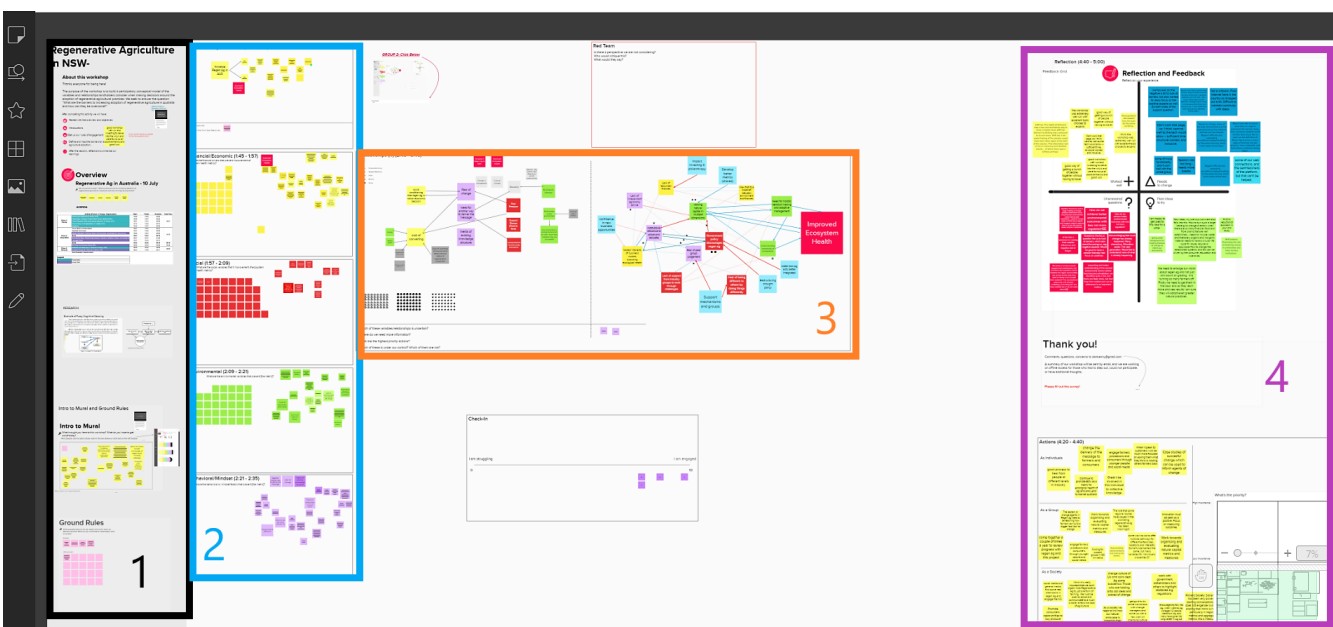

**Figure 2.** A screenshot of the structure we created on MURAL for each of the four stages of the workshop: (1) plenary, (2) elicitation, (3) modelling, and (4) debrief.

We put together a facilitation team to work with each stakeholder group (participants were randomly split into two groups for the elicitation and modelling stages before reconvening for the debrief). Each facilitation 'team' had a facilitator (1 an expert, 1 a PhD student), a modeller (expert), and a support person (student), and the entire workshop was video-recorded (with the informed consent of participants). The facilitator was in charge of leading participants through the process of building the FCM, solving any issues as they arose. The modeller was responsible for drawing on MURAL for the modelling phase (*at the direction of participants*) and creating 'stickies' for participants who had trouble accessing MURAL. The support person kept an eye on the chat and the video to make sure participants had the chance to participate. They asked clarifying questions when necessary. The research team trialed the process a month beforehand with a group of postgraduate students and academics to gain familiarity with the virtual methods and to 'stress test' the virtual approach. We also provided participants with tutorial videos on how to use Zoom and MURAL in the week leading up the workshop to level set for digital skills during the workshop.

### 3.2.1. Plenary

We developed pre-defined templates in MURAL to provide a sequence of activities for the group to work through. The plenary session lasted 30 min and started with introductions from participants and the research team. This included introductions to the overall research of the lead facilitator, an overview of the work of the Mulloon Institute, a presentation on the modelling process and the concept of cognitive biases, and a tutorial activity to make sure everyone felt comfortable using MURAL's features (e.g., adding a 'sticky' note, commenting on other notes, and voting). In the first activity, we discussed our shared purpose and setting the goal and modelling 'metric' that would be the focus of the subsequent elicitation and modelling stages. Considering that the theme of the workshop was on the barriers to adoption of RegenAg, the discussion centred around what 'metric' or indicator might best reflect the success or failure of this effort. We set aside 15 min for this activity, considering we had already introduced initial ideas for the metric (the

number of farmers practicing RegenAg and/or the number of hectares under RegenAg production) over email prior to meeting. As a group, we ended up needing more time for this activity, as many in the group had different ideas about what the metric should be, and, quite understandably in retrospect, more time was needed to explain the purpose of this 'metric' in defining the process of the workshop. This initial framing of possible metrics may have also introduced 'bias' in pre-determining what was considered; we chose to do so in the interest of time and a desire to engage in the modelling activity. Ultimately, our group decided that '*Improved Ecosystem Health*' was the ultimate 'metric' or outcome of interest that RegenAg sought to promote. Participants were then told that the goal of the following activity (the elicitation phase) was to unpack key variables that directly or indirectly contribute to the state of this outcome. Then, during the modelling phase, we would set out to establish how these variables might be interrelated and to identify a portfolio of levers that could exert a positive effect on key variables within the system.

### 3.2.2. Elicitation

Armed with the notion of '*Improved Ecosystem Health*' as the outcome of interest for the system, we used Zoom's breakout room feature to divide participants into two groups (group 1= six participants, group 2= five participants) to build two FCMs. Each group then moved through the process of brainstorming and ranking of causal 'factors' or 'variables' that might contribute to or hinder '*Improved Ecosystem Health*' under four categories that were pre-established by the research team: (1) Economic/Financial, (2) Environmental, (3) Social and (4) Behavioral.

Under each category, the group's participants were given time to write down their thoughts on what the key factors either driving or hindering the adoption of RegenAg are, which were based on their knowledge, experiences, beliefs and perspectives. We did not impose a limit on how many factors each participant should contribute, and they completed this activity on their own—with little to no discussion—to avoid groupthink [59]. We then proceeded to discuss, as a group, what each participant's factor contributions meant, with the aim of casting a vote on the top three factors. For each category of factors, we allocated 5 min to the individual elicitation of factors, 10 min to group deliberation, and 5 min for a polling activity (participants were allocated 3 votes to allocate to the group's sticky notes). We repeated the process for each of the four categories, ending with twelve factors after 80 min that progressed to the modelling phase.

### 3.2.3. Modelling

During the modelling phase (which was originally allocated a total of 60 min, but we had closer to 45 min), and taking cues from participants' views expressed through an open discussion, the facilitating team began to draw connections (*at the direction of participants*) between the top twelve factors, establishing positive and negative relationships and how 'strong' these relationships were between the factors—i.e., their polarity. Participants decided how these factors related to each other and how they contributed to our ultimate 'metric' of '*Improved Ecosystem Health*' (Figure 3). The role of the facilitators during this phase of the participatory process was to draw the connections between factors as decided and directed by participants, inquire as to their polarity and strength, and to ask clarifying or prompting questions about why that relationship existed and if there were other factors to consider. One person's role was to keep the conversation moving, and one of the other facilitators primarily handled the technical aspects of the FCM modelling, primarily in drawing the connections. An additional support person monitored the chat room on Zoom for any nonverbal contributions or if the facilitator missed a participant eager to say something.

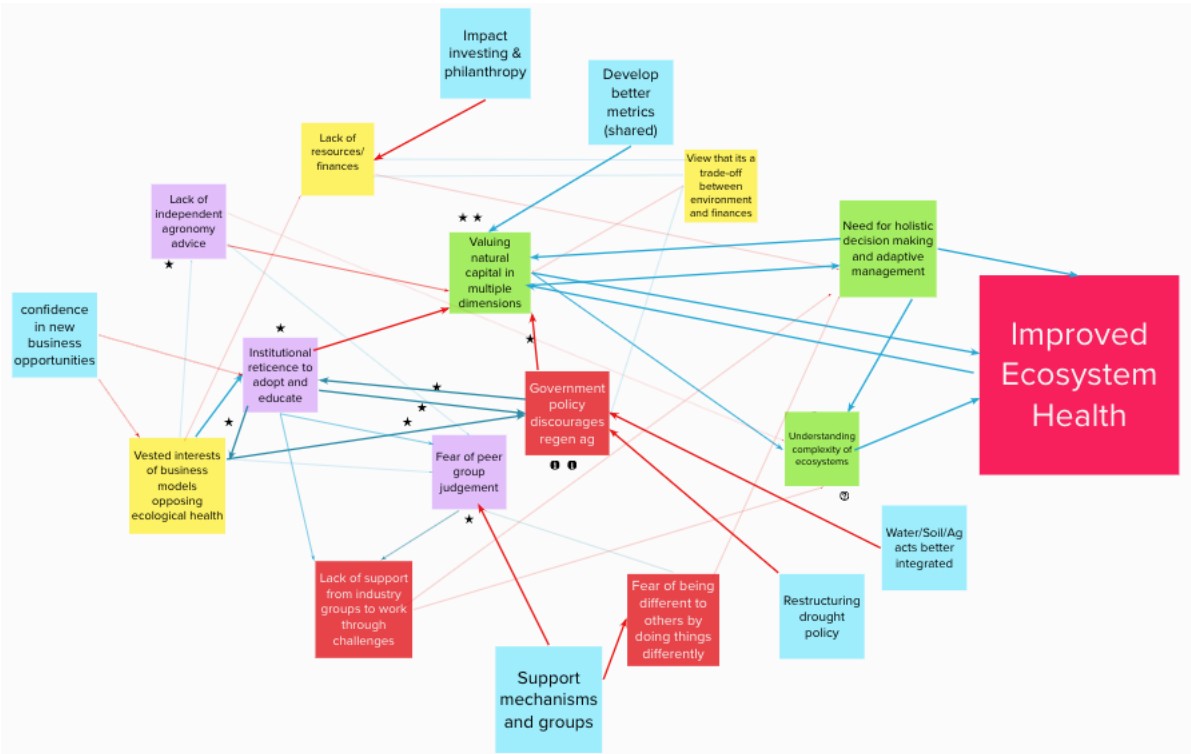

**Figure 3.** A screenshot of the initial, complete FCM built by Group 2 during the modelling phase in MURAL. Green stickies were environmental, yellow stickies were economic/financial, red stickies were social, purple stickies were behavioral, and blue stickies were 'levers' or actions that could be taken to influence the system. Our metric, '*Improved Ecosystem Health*', was the measure we sought to influence and understand, as a broader proxy for the success of RegenAg. Arrows indicated causal relationships, with bold arrows indicating 'strong' relationships, and pale arrows indicating 'weak' relationships. Red arrows were negative or 'balancing' relationships, and blue arrows were positive or 'reinforcing' relationships.

In planning the workshop, we set aside time for when the group finished building the mental model map (i.e., identified all relevant causal relationships between factors for the four categories and established their polarities); they could also look to identify and add 'levers' to the map, actions or policies that might be capable of shifting key 'factors' to move towards a better state for the chosen 'metric'. For example, Group 2 identified several levers, including '*Restructuring Drought Policy*' to affect '*Gov't Policy Discouraging Regen Ag*' and '*Support Mechanisms and Groups*' to address '*Fear of Peer Group Judgment*' and '*Fear of Being Different to Others*'. In short, 'levers' were ways participants identified an ability to 'shift' or 'transform' a factor within the model and thereby alter the system in the hopes of moving closer to the metric of '*Improved Ecosystem Health*'.

One of our groups (Group 2) made it through this stage, building a complete 'map', while the other (Group 1) struggled to complete even drawing connections between the 12 identified factors. In the former, the group did a much better job of staying on task (possibly due to the experience of the lead facilitator, who was practiced in leading PM), while the latter struggled through the initial phase of the exercise in eliciting and ranking factors (possibly due to the inexperience of the lead facilitator, or greater confusion over the purpose of the exercise, or both).

3.2.4. Debrief

The workshop closed with a 'debrief' session consisting of two parts. In part 1 (20 min), the two groups were brought together to share and present their FCMs, focusing on the similarities and differences between the two. At this stage, we were not seeking to combine

the models but rather to compare them. After discussing the two FCMs, we set aside 30 min for the whole group of participants to reflect on actions that could improve the adoption of RegenAg (10 min) and for the group to reflect on their experience of the workshop (20 min). We divided the actions discussion into those that could be taken as individuals, as a group, and as a society, with time given for participants to put down their individual thoughts on their own (no discussion) before coming together to briefly discuss these as a group. Reflection on the workshop was guided by a template available on MURAL (see figure below), dividing quadrants between "What worked well?", "What needs to change?", "What are new ideas to try (for next time)?", and "What are the unanswered questions?" Participants were given time to put down their thoughts and discuss section by section.

### 3.2.5. Follow-Up

Follow-up interviews with participants were conducted to evaluate their experience of the workshop and to provide space for feedback (on the model or the process) that they were unable to provide during the workshop. The aim of this follow-up process was to combine the FCMs developed by the two groups into one model by cross-checking and comparing them to find common variables. We also re-visited the recordings and transcripts of the workshop to determine what possible connections could be added (similar to the Rich Elicitation Approach [60]). The combined model was digitised into the Mentalmodeler.org FCM analysis software and sent to participants via email for approval and feedback [60] with a full record of the changes made for the stakeholders to validate.

As a result of this follow-up process, we also identified key 'narratives' present in the FCM using qualitative analysis. Although the relatively small sample size of our study (15 stakeholders) limited our ability to draw statistical inferences from the FCM, it enabled us to work closely with stakeholders in the process of validating and extracting insights from it. We used thematic analysis to identify the emergent themes and patterns that arose from the workshop [61] to upscale individual experiences and perspectives to structural themes and patterns [62]. These themes and patterns ultimately became our narratives. Narratives have been previously used to communicate the results of FCM [63], but they are not widely used in PM, despite their suitability for communicating complexity [64] and their relevance to environmental issues [65]. To identify the narratives, we searched for the structural barriers facing the adoption of RegenAg, drawing on several iterations of a synthesis of the model variables and relationships, the audio recordings, transcripts of the workshop discussions, and submissions on the workshop pre and post-questionnaires. We followed the approach laid out in [61], first by analysing workshop transcripts, questionnaire responses, and re-visiting the recordings, before generating initial themes on the transcript, revisiting those themes (in combination with looking at the model for the relevant variables and relationships), and then reviewing those themes, written as narratives, with stakeholders to ensure their validity. Our inductive coding process did not look at frequency patterns, instead looking for 'themes' at the passage level of the transcript and questionnaire responses *and* its respective grounding in the variables and relationships present in the model.

As our stakeholders were not experienced modellers, we deliberately used narratives to communicate and interpret the model results for participants, similar to Eakin et al. [63], but using a template of our own (available in Results: 'Meso-scale analysis: FCM Narratives'). Our template highlights the story, the actors, and the role of the narrative within the model and potential solutions. Five narrative templates were presented to participants to validate and/or propose any changes, to the model and/or to the narratives themselves (see Section 4.1.3).

### 3.3. Data Analysis

In analysing Fuzzy Cognitive Maps, Ref. [52] lay out the steps to move towards analysing a social model, which includes determining an adequate sample size, using

graph theory to analyse the structure of the models, condensing the models for comparison, and then using neural network computation to analyse outcomes and simulate different policy options. After transcribing the aggregated FCM into the MentalModeler software (www.mentalmodeler.org), we were able to calculate the following statistics [66]:

- Total number of variables;
- Total number of connections;
- The network 'density', as the actual number of connections divided by the number of connections possible in the 'map' (i.e., if all variables were connected to each other, that would be a density of 1);
- The average connections per variable;
- Complexity score, as the ratio of receiver variables to transmitter variables;
- Centrality rankings as a proxy for the most 'influential' variables, which depends on the number and strength of the connections attached to a variable. The higher a variable's centrality, the more influence it has on the 'map' when it changes.

In examining the maps, it is important to identify transmitter, receiver, and central variables. The 'centrality' of a variable is determined by the number of relationships and the cumulative weight of those relationships coming in and going out; the higher that number is, the more important that variable is to the feedbacks of the system [57]. Transmitter variables (or forcing functions) have a lot of relationships going out and none coming in, while receiver variables take in the relationships of other variables and send none out [46]. Identifying and labeling these variables in an FCM can help generate insight into the way agents view their system; for example (taken from [46]):

> *"Local people and hunters have more transmitter variables in their maps than NGO personnel [did in their maps]. This indicates that local people and hunters see themselves and the Uluabat Lake ecosystem as being under outside control and dependent on outside forces."*

In all FCM exercises, it is important to remember that complexity is not the ultimate goal; the aim is for the model to be a useful representation of reality [52], whether there are many variables or a few. The number of variables in a map does not determine its success, and therefore, we encourage the approach of keeping a model as simple as possible to solve a particular problem, and no simpler [57].

## 4. Results

### 4.1. Macro-, Meso- and Micro-Scales of FCM

The model created in the aftermath of the workshop (Figure 4) reflected a collaboration of individual mental models, and we devised a process to analyse this at three scales: the macro (a comparison between all of the networks and their global characteristics, see Section 4.1.1), the meso (an analysis of the communities, sub-networks, and narratives present in the model, see Section 4.1.3), and the micro (an analysis of the most relevant 'variables', see Section 4.1.2). With each of these 'lenses', it is possible to extract and condense the critical information of the 'map' to answer key questions. For example, what is central to each network? How do you compare between networks in structural and statistical terms? Which 'variables' from each category (financial, social, environmental, and behavioral) are the most central to the network? Where are the densest 'zones' of the network? As we show below, using the different scales of the FCM can further illuminate the principal networks, variables, and narratives present in the system as identified (explicitly or implicitly) by workshop stakeholders.

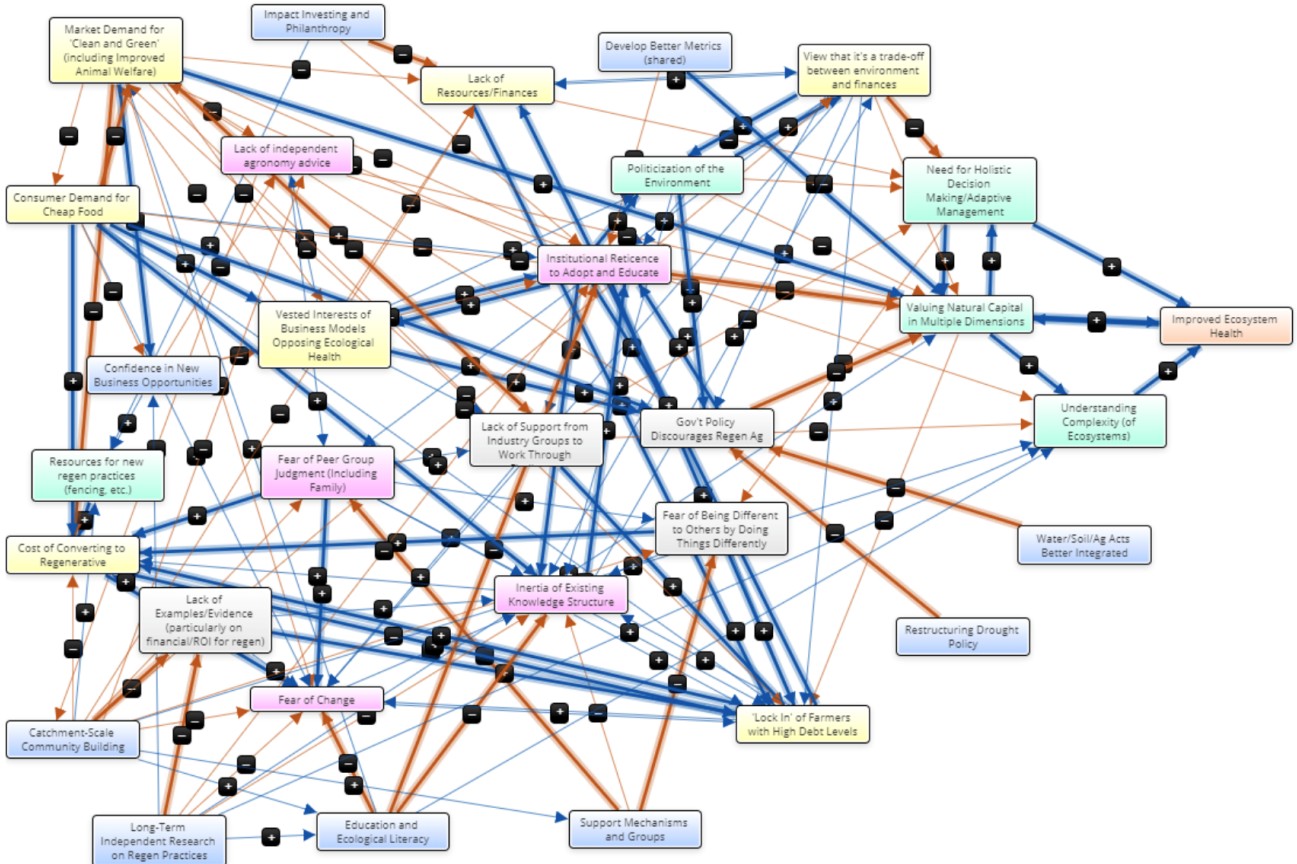

**Figure 4.** The combined FCM elicited from workshop participants. We used different colors to reflect the different variable categories, similar to what we had used in MURAL (green = environmental, yellow = financial/economic, gray = social, and pink = behavioral), while our 'levers' or actions we could take were in blue. Our metric, '*Improved Ecosystem Health*' in orange, was the measure we sought to influence and understand, as a broader proxy for the success of RegenAg. Arrows indicate causal relationships. The color of the arrows indicated their polarity (red arrows were negative or 'balancing' relationships, and blue arrows were positive or 'reinforcing' relationships), and the width of the arrow indicated the strength of the relationship (thick were 'strong' relationships, thin were 'weak' relationships).

#### 4.1.1. Macro-Scale Analysis: The System

The final, aggregated FCM (Figure 4, Table 1) comprised 31 concepts, with 141 relationships. This included five transmitter components (levers: '*Impact Investing and Philanthropy*', '*Develop Better Metrics*', '*Restructuring Drought Policy*', '*Water/Soil/Ag Acts Better Integrated*' and '*Long-Term Independent Research*'), and only one receiver component (the goal of the system, '*Improved Ecosystem Health*'). The five most 'central' variables were (in order) '*Institutional Reticence to Adopt and Educate*', '*Valuing Natural Capital in Multiple Dimensions*', '*Cost of Converting to Regenerative*', '*Gov't Policy Discourages Regen Ag*', and '*Lock In of Farmers with High Debt Levels*".

**Table 1.** FCM results (network statistics).

| Statistic | Total | Interpretation/Insight |
|---|---|---|
| **Variables** | 31 | Each iteration of 'modelling' added more variables as stakeholders further appreciated the complexity of the system, and the interconnection of various forces, including economic, social, environmental, and behavioral, that are present and interacting in this system. |
| **Connections** | 140 | With the addition of 'variables', the number of connections greatly increased from the first iteration of the model to the current version. This is reflected in the Connections Per Component, which roughly doubled from 2 to 4.5 from the initial workshop to the final iteration of the model. A greater number of connections could represent an increasing recognition of the interconnectedness of the system. Those increasing dependencies could make for a challenge, in the 'wicked complexity' [67], of not being able to isolate variables within the system, but they also could present an opportunity in that the right actions can have far-reaching effects in the system. It is also possible that the increase in connections is a result of more time given to participants in subsequent follow-ups, in contrast to our original modelling protocol which was designed to deliberately keep things simple. |
| **Network Density** | 0.15 | Our network density was quite low, although we did see an increase from the first iteration of the model to the final version. As the actual number of connections divided by the number of connections possible in the 'map', the more connected the map and the variables within became, the higher the density. We expected the increase in the number of connections as participants grew more comfortable with the modelling process and had more time to think of them in the follow-up outside the constraints of the workshop. The increase in connections reflects the 'wicked complexity' [67] of this system as an interconnected web of financial, social, environmental, and behavioral drivers. |
| **Connections per Component** | 4.51 | From the initial model to the final iteration, the Connections Per Component roughly doubled from 2 to 4.5. The majority of the connections in the map are 'positive' in their influence, so this increase in connections could, without the presence of balancing 'negative' connections, further spiral the system deeper into the conventional agricultural paradigm and, according to the model, decreased ecosystem health. However, as it is depicted in the system, were the trend in conventional agricultural dominance to reverse, those same 'positive' feedbacks present could spiral the system upwards towards better ecosystem health. This seems unlikely given the institutional and slow-moving nature of government as a key player in this system, but it is possible. Meanwhile, an increase in the number of negative feedbacks in the system, particularly around 'central variables', means that spiralling or rapid transitions of any kind (positive or negative) are less likely to occur as there are more 'balancing' relationships present to counteract the 'reinforcing' relationships. |
| **Complexity Score** | 0.20 | This is a low score, as we only had one receiver component, our metric for the system of '*Improved Ecosystem Health*'. This 'score' is specific in the way it defines complexity as it perceives a model to be less complex "when many transmitters are represented with only a few outcomes (receiver variables) of those pressures represented" [43]. More 'outcomes' could be added to the model, but the process we used was quite structured in using one metric to guide and narrow the focus. |
| **Transmitter Components** | 5 | The relatively low number of transmitter variables is likely a product of our approach to building the model. All of these components were 'levers', variables added to reflect actions that could be taken to influence the system. In that sense, it makes sense they have only outgoing arrows, designed as they are to 'impact' the system. In further iterations of the model, it would be interesting to see how other variables, especially other levers, might integrate with these identified variables, as it happened with other variables (ex; *Long Term Independent Research* has a positive effect on *Education and Ecological Literacy*). |
| **Receiver Components** | 1 | This was our goal and metric for the system, '*Improved Ecosystem Health*' that we established at the beginning of the exercise. As such, changes in the system, in theory, should affect this metric, for good or ill. It is likely other metrics exist and could be used to monitor different parts of the system. Focusing on one metric was an intentional choice to narrow the discussion for our workshop. |
| **Ordinary Components** | 25 | The majority of our variables were ordinary components, meaning they were variables with incoming and outgoing connections. As noted by [43], this demonstrates the "significant interlinkages and influences between system components", which is potentially a sign of further complexity in the system. It is unsurprising how interwoven this system is, as agriculture, and RegenAg in particular, is a product of "multiple, simultaneous and inter-connected ecological, economic and social pressures" [9]. |

### 4.1.2. Micro-Scale Analysis: Honing in on Variables

**Centrality.** 'Centrality' in an FCM serves as a proxy for the most 'influential' variables in a given network (Table 2). Each variable's centrality score depends on the number and strength of the connections attached to a variable. The higher a variable's centrality, the more influence it has on the 'map' when it changes, which is a result of the combination of the number of relationships and the cumulative weight of those relationships coming into and going out from that variable [57].

**Table 2.** FCM results (centrality).

| Component | Centrality |
|---|---|
| Institutional Reticence to Adopt and Educate | 10.1 |
| Valuing Natural Capital in Multiple Dimensions | 7.9 |
| Cost of Converting to Regenerative | 7.5 |
| Government Policy Discourages RegenAg | 7.4 |
| 'Lock In' of Farmers with High Debt Levels | 7.3 |
| Market Demand for 'Clean and Green' | 5.7 |
| Consumer Demand for Cheap Food | 5.2 |
| Inertia of Existing Knowledge Structure | 4.8 |
| Vested Interests of Business Models | 4.2 |
| Politicisation of the Environment | 4.1 |
| Need for Holistic Decision Making/Adaptive Management | 4.0 |

Measures of 'centrality' can provide insight into key traps and pain points in the system, potentially as areas to focus on and 'leverage points' of opportunity for change [68]. For example, institutions, mainly the government, play a large role in this system, as the most central variable is '*Institutional Reticence to Adopt and Educate*', with '*Govt Policy Discourages Regen Ag*', '*Inertia of Existing Knowledge Structure*', and '*Politicisation of the Environment*' all making the top ten. This suggests that the dominant paradigm of government policy is one supportive of conventional agriculture, which is borne out by the role conventional agriculture plays in the Australian economy and the relatively low percentage of Australian agriculture belonging to RegenAg or other alternative measures. This has the potential to be a reinforcing feedback loop that solidifies lock-in to traditional agricultural practices, particularly considering the influence of the Market ('*Consumer Demand for Cheap Food*') and Business ('*Vested Interests of Business Models*'). Without policies or actions to provide a balancing relationship (reflected by the red arrows and largely present from the 'levers' in the map), these variables are all connected by positive relationships, with an increase in one leading to an increase in another without an obvious incentive to change. This perception among stakeholders is striking, although it is perhaps not surprising given the number of stakeholders in the workshop who self-identified as 'pioneers' or 'mavericks' during the modelling process. Their position on the 'outside' as early adopters and advocates of RegenAg may give them more insight into how the paradigm of conventional agriculture supports and intensifies (through positive relationships) its own structures.

According to workshop participants, business and industry is also largely arrayed against RegenAg ('*Vested Interests of Business Models*', '*Consumer Demand for Cheap Food*', '*Lack of Support from Industry Groups*'), although they also noted the opportunities within that sector ('*Market Demand for Clean and Green*'). The connections between '*Consumer Demand for Cheap Food*', '*Vested Interests of Business Models Opposing Ecological Health*', and '*Gov't Policy Discouraging Regen Ag*' had strong positive arrows between them, suggesting a reinforcing system that is difficult for RegenAg to 'break into' without serious policy or business investment and intervention.

Possible ways to intervene in this system were identified by the 'levers', the blue variables, which are either transmitter variables (outgoing connections only), or loosely and weakly influenced by other levers, such as '*Long-Term Independent Research*' having an effect on '*Education and Ecological Literacy*'. None of these levers, which can also be seen as interventions in the system, rank highly in centrality. This is unsurprising, as they were added last as inputs into the system, limiting their connections and therefore their centrality, but it is also worth noting, as it also may reflect the difficulties of influencing this system with so much 'reticence' and 'inertia' ingrained. This would reiterate the need to find balancing relationships within the system, starting with '*Valuing Natural Capital in Multiple Dimensions*'.

'*Valuing Natural Capital in Multiple Dimensions*' scores highly on centrality, which is a reflection of the number of strong connections it primarily receives. This was a key

variable of focus for participants, with strong ties to a '*Need for Holistic Decision-Making*' and '*Understanding Complexity of Ecosystems*' as environmental variables capable of directly improving the metric of '*Improved Ecosystem Health*'. Others in agriculture, and more broadly in sustainability [69,70], have called for a valuing of natural capital, and it stands to reason this would be a vital issue for stakeholders as they seek to place a financial value on the often ignored positive externalities of RegenAg. Determining where and how to influence' *Valuing Natural Capital in Multiple Dimensions*', the second most central variable on the map, would be a vital first step for those seeking to increase the adoption of RegenAg.

Such efforts to value natural capital may also help to address two of the closely related variables that also score highly on Centrality: '*Lock-In of Farmers with High Debt Levels*' and the '*Cost of Converting to Regenerative*'. For participants, these variables reflected the difficulties, as one put it, of "going green when you're in the red". Transitioning to RegenAg often requires a high upfront cost either in additional resources (ex, the cost of fencing to move to cell grazing) or in reduced income by shifting away from high-production farming. These additional expenses may or may not reflect reality, particularly when considering the return on investment and resilience offered by many RegenAg practices, but the 'perception' of the expense seems to be important for those considering a transition. As many farmers are already in debt, these remain significant challenges, even when ignoring the additional social costs and mental strains of making such a change while dealing with judgment or pressure from peers and family. The additional variables of the '*Lack of Resources/Finances*' and the '*View that It's a Trade-Off Between Environment and Finances*' lend further support to the significance of this barrier facing those seeking to transition to RegenAg. Understanding the variety of factors affecting this particular barrier, while complex, is possible due to the visual and interconnected nature of the FCM.

**Complexity.** Over the course of the model iterations, the average number of connections for each variable increased from 2.5 to 4.5, which was likely due to participants feeling more comfortable expressing how interconnected the system was and the fact that they were given more time to reflect on and internalise the knowledge captured in the map. These changes in the FCM demonstrated a greater understanding of the connectedness of the system, reflecting the complexity of the issue and showing evidence for social learning [43,71,72]. It is possible that given more time, stakeholders would continue to identify new connections, but there is a risk of diminishing returns, as more connections does not always lead to greater understanding.

It is also worth noting that despite the increase in the average connections reflecting a greater complexity to the system, this was not reflected in the Complexity Score, which was defined as a function of transmitters to receivers, which was quite low at 0.2 (on a score ranging from 0 to 1, with 1 being high complexity) [43]. We were not concerned by this result, as this was a product of the process we used, beginning with a singular metric, '*Improved Ecosystem Health*', which was our sole receiver variable and the measure for which we sought the system to work towards improving. Other authors have noted this score might not necessarily negate the complexity of the model, as the lack of receiver components "could in fact be a sign of a complex model that shows significant interlinkages and influences between system components" [43].

Furthermore, workshop participants demonstrated an understanding of complexity and systems thinking during the workshops, which may have been self-selected among RegenAg practitioners who have to account for the effects of biodynamic influences on their economic and social activity of agriculture. It is still noteworthy, as they noted the reinforcing nature of a number of institutional barriers and policy barriers that were identified as a part of the workshop. For example, participants noted the policy 'triangle of death', which was a reinforcing loop between '*Vested Interests of Business Models*', '*Institutional Reticence to Adopt and Educate*' and '*Gov't Discourages Regen Ag*'.

> "*In a negative sense, we were focused on a little triangle of interactions between the yellow one on the bottom left, the vested interests, linking up to institutional reticence and linking across to government policy. And that little circle travels its own kind of*

*thing in a vortex to the bottom and takes us away from what we need to do in valuing natural capital and improving ecosystem health. So those vested interests... they were a couple of key factors."*

While not explicitly stated as such, this is evidence of systems thinking in identifying the microstructures or systems motifs (in this case, a moderated effect motif) present in the network, linking causality that accelerated the effect that 'Vested Interest' and 'Gov't Policy' both have on 'Institutional Reticence' [73,74]. It is encouraging that RegenAg practitioners were aware of these feedbacks within the system, as previous studies had shown that familiarity with systems thinking among participants was not necessarily reflected in any subsequent 'model' [73,75], and systems thinking has long been linked with positive outcomes in sustainability and the ability to improve decision making in SES [44,73,74,76–78].

4.1.3. Meso-Scale Analysis: FCM Narratives

'Narratives' have been used as a part of qualitative research in psychology, anthropology, sociology, health research, and climate and energy research [65,79]. It can be defined as "collecting and analyzing the accounts people tell to describe experiences and offer interpretation" [79]. In the aftermath of the construction of the FCM, we analysed the initial model results, the recordings of the workshops, and the interviews with participants. As a result of this process, we identified five key 'narratives' in the FCM, and we used these to communicate the model results to participants, asking them to offer feedback on the validity of the narratives and on their implications. Each narrative consists of the following:

- **The Story:** a brief description of what the narrative is.
- **The Actors:** identifying who the key players might be in such a story.
- **The Model:** what implications this story has for the model (in our case, the representation of the RegenAg system using FCM), both in how it is currently reflected in the model and what might need to change accordingly.
- **The Solutions:** if this story is true, what solutions or actions are needed to address the narrative in seeking to increase adoption of RegenAg practices.

We see these narratives as a means to establish an ongoing conversation between participants, especially as some participants reached out to the research team after the workshop to share their own narratives. The methodologies used to elicit and co-construct narratives from FCM workshops will be the subject of a forthcoming paper by the authors.

By focusing on extracting narratives from the FCM—as opposed to the more traditional quantitative analysis)—we sought to avoid the criticism and the challenge of the 'black box' of modelling (obscuring the workings of the model from stakeholders) by laying out what patterns were present (in the form of stories) and clearly identifying where variables and relationships in the model supported this story or where it might need to change to better reflect this pattern. Furthermore, the narrative construct identifies key actors and solutions as a way to put this narrative 'into practice' and to empower stakeholders to act. It is a similar approach to Scenario Analysis [75] that often works hand in hand with PM, but we used these narratives primarily as a communication tool. The iterative nature of moving from narrative to model, and model to narrative, especially with a visual component, allows for the elicitation and refinement of implicit mental models made explicit by our PM workshop. Multiple narratives can (and probably should be) present, helping to convey the complexity of these issues without being overwhelming. Anecdotally, after the workshop one participant responded, unprompted, with a narrative of their own (which became Narrative 5), making reference to the model, the variables, and relationships, attesting to the simplicity and efficacy of narratives as a communication device between the research team and participants.

> *"...I gave some thought to all that you had written and concluded that the most constructive way for me to respond to you was to simply create a fourth narrative in the attached document..."*

In the Discussion section below, we argue for the systematic use of these narratives as one way to address the challenge of communicating complexity during PM processes. The five narratives elicited in our PM workshop are presented in Tables 3–7.

**Table 3.** Narrative 1: Government First.

| Narrative | Component | Description |
|---|---|---|
| **Government First** | Story | The culture and current paradigm is so enshrined in society that only the government has the resources and the ability to break us out of it. It is their job to protect the environment and future generations and they must act and do so quickly. Their investment in and provision of incentives for transitioning to RegenAg is the first step in creating a spill-on effect to the rest of the system. |
| | Actors | The Australian Government/The Prime Minister and Cabinet (particularly Ministers of: Agriculture, Drought and Emergency Management, Environment, Education)/Voting Public |
| | Model | *'Government Policy Discourages Regen Ag'* and *'Institutional Reticence to Adopt and Educate'* are two factors at the centre of the model, and both are currently in the top 5 'most central' and influential variables. This is a result of the number of strong, causal arrows emerging from and being received by these factors. Should this narrative be true, even more of these links will need to emerge from these 'variables' to create effects on the rest of the system as 'forcing' mechanisms. The pressure would then be to create relationships that positively influence these key variables to 'encourage' more of a transition or transformation to RegenAg. |
| | Solutions | If 'Big Government' is the problem, then 'Big Government' must be a part of the solution, although this includes the federal and state government. While noting the effect the individual voters and media have on the government, if this narrative is true, a drastic reform at the level of Federal Government policy is needed. More incentives need to be provided for a switch to RegenAg and could show up in a reform of drought policy or an integration of water and soil acts, perhaps more in line with watershed boundaries as opposed to arbitrary political lines. Workshop participants noted that not only does this commitment need to be significant, it also needs to be 'long-term' in order to align with the cycles of natural capital and to "give confidence to land managers, industry, educational institutions, NGOs, and the broader public." Part of that effort could then include greater efforts to inform the voting public of these issues and/or actively push Parliament to embrace policies benefiting regenerative agriculture by directly lobbying the relevant Departments and Ministers. |

**Table 4.** Narrative 2: The Market Matters

| Narrative | Component | Description |
|---|---|---|
| **The Market Matters** | Story | The combination of 'Consumer Demand for Cheap Food' and 'Vested Business Interests', along with the surrounding infrastructure, can keep conventional agriculture in place by creating a system that seems to race to the bottom. As noted by one participant, the "consumer demand/expectations for 'brandless' cheap food commodities [is a] major hindrance" to the adoption of regenerative agricultural practices, as they tend to carry a higher upfront cost and often necessitate premium pricing as a result. Shifting this demand to food more aligned with holistic and regenerative practices puts pressure on businesses and government to incentivise those practices further and provide the structures and policies needed to produce at scale. |
| | Actors | Australian consumers/Woolworths, Coles, and other supermarkets/Agribusiness/Department of Agriculture/Banks and Financial Institutions |
| | Model | Much of this story is reflected in the upper left corner, with 'Consumer Demand for Cheap Food', 'Vested Business Interests' and 'Lack of Support from Industry Groups' all playing a key role in keeping the current paradigm focused on conventional production to meet the needs of the market. If this narrative were true, the connections between these three variables would be strong, and would further tie into 'Institutional Reticence', 'Gov't Policy Discourages Regen Ag', *'Lack of Resources/Finances'*, and *'Cost of Converting to Regenerative'* to lock in market control and limit regenerative to a niche category. To change it, this reinforcing system would need to be broken. |
| | Solutions | Find ways to increase consumer demand for products of RegenAg (affecting 'Consumer Demand for Cheap Food' and 'Market Demand for 'Clean and Green''), which could happen in a number of ways: (a) Provide government incentives to subsidise the cost of regenerative products, either in out of the gate packaging and production, or in reducing the high upfront costs needed to switch to regenerative. (b) Create regional processing and distribution centres in high agricultural areas devoted to regenerative products and lowering costs by producing at scale. (c) Incentivise supermarkets to carry regenerative products either at lower prices or in high-value locations in stores to encourage more sales. (d) Increase funding to marketing and advertising to craft a more compelling narrative for regenerative products to direct-sell to consumers. |

**Table 5.** Narrative 3: Pressured Communities.

| Narrative | Component | Description |
|---|---|---|
| **Pressured Communities** | Story | Our communities have been conditioned to feel conventional agriculture is the only way, and this is present in our interactions with family members, neighbors, and peers in the agricultural world. As noted by one stakeholder, the supporting structures around agriculture (banks, agronomists, certain industry groups) also "have a lot invested in conventional farming", which strengthens this connection. Unless we can actively promote supportive mentors, community champions, community groups, and a solid evidence base, people will continue to avoid transitioning to practices viewed as 'unconventional', even 'radical'. |
| | Actors | Landcare/Land Services (LLS)/Local councils/Banks (Bank managers and accountants)/ Agronomist groups/RegenAg practitioners and trainers/Individual landholders and farmers/Non-profit organisations |
| | Model | Currently, the fear-based trifecta, 'Fear of Peer Group Judgment', 'Fear of Change' and 'Fear of Being Different to Others', play a central role here (which is similar to 'Start with People', Narrative 4). However they combine, with the 'Lack of Examples/Evidence' and the 'Cost of Converting' and 'Lock In' of Farmers with High Debt Levels' to create a risky proposition of upsetting social norms with the possibility of little to no pay-off. If this were the dominant narrative, these factors would need to be more connected to the 'Vested Interests of Business', as this provides further disincentives and possesses a stronger connection to the 'View that Regen Ag is a trade-off between environment and finances' as this is a fundamental assumption of many who resist transition to RegenAg. These would have to be far more interconnected with the rest of the map, with 'Support Mechanisms and Groups' and '*Long-Term Independent Research* on Regen Practices' being pushed as a way to address these challenges. |
| | Solutions | The policy should centre on 'normalising uptake' of RegenAg practices to remove any social stigma that comes from such a transition or practice. Being able to point to indicators of success, or cultural capital [18,80,81] for regenerative farmers, such as increased income or production, can help shield such farmers from criticism. Therefore, building up evidence and case studies to complement these transitions, ideally over the long term, can help. In addition, identifying and working with local and community champions, which could include members of local councils, NGOs, or fellow farmers, could present additional social visibility and support. Providing training and support and building "communities of practitioners and networks of conversation" can also assist and can span across regions due to the access and ease of the internet and social media channels. |

**Table 6.** Narrative 4: Start with People.

| Narrative | Component | Description |
|---|---|---|
| **Start with People** | Story | We have to start from the ground up in creating a cultural change, by capturing the hearts and minds of farmers through conversations, education, and outreach. This is where conversations and dialogue need to proceed both in understanding where individual circumstances work against a transition to RegenAg and in tailoring messaging to highlight if, where, and how practices of RegenAg might better align with the values and beliefs of farmers considering a transition. |
| | Actors | CSIRO/Universities/RegenAg practitioners and trainers (including holistic management, landscape rehydration, and other areas)/Non-profit and research organisations (such as The Mulloon Institute and Soils for Life)/Individual landholders and farmers |
| | Model | This narrative puts the various 'fear' variables as the central focus, along with the 'Inertia of the Existing Knowledge Structure'. If this narrative were true, this would also require recognising how deeply rooted the trifecta of fear variables, namely 'fear of change', is in individual minds, increasing the number of arrows emerging from this space, many of them weak, but with deep roots throughout the system. As a result, '*Education and Ecological Literacy*' and 'Understanding Ecological Complexity' would need to play a much greater role in addressing these influences. |
| | Solutions | Appropriate solutions would need to address the fear that underlies much of the social and institutional resistance. Education and outreach would be the centrepiece of this effort to have conversations and dialogue with the aims of transforming the hearts of minds of the agricultural industry, highlighting the "hope, dreams, and aspirations" of "leaving the land in better shape for the next generation". This would also include better marketing and targeting of consumers to have them switch to products of RegenAg. However, it is likely this scale of change is likely to be long term, as noted by participants, likely decades. |

**Table 7.** Narrative 5: Community by Community.

| Narrative | Component | Description |
|---|---|---|
| **Community by Community** | Story | As agriculture is often an area confined by ecological boundaries, we have to identify potential communities of farmers within watershed areas and provide incentives for individuals to change and for communities to collaborate. This is similar to Narrative 3 but instead focuses on an organisation-led charge to benefit from the emergence of social, economic, and environmental outcomes that arrive from mutual support and collaboration on a conversion to RegenAg. Farmers need support from each other and from experts to complete this change successfully. The challenge is to find and/or create the good reasons that the members of a potential community will need to convert to RegenAg. That may include new marketing opportunities, or changes at the watershed scale allowing for improved production, or new social opportunities resulting from collaboration. Ideally, a permanent organisation must be set up to undertake the work of identifying suitable communities, convincing the community to join, and overseeing the process of this change. The role of this organisation is to marshal the expertise and resources required to totally transform existing farming operations into truly Regenerative Agricultural farms in such a way that the farmers involved achieve better outcomes and significantly improve their quality of life in ways that are congruent with their values. |
| | Actors | Community-led organisations or non-profits/RegenAg practitioners and trainers (including holistic management and landscape rehydration)/Business consultants (including people experienced in community organisation and decision making, marketing and sales, logistics, technology and finance)/Farmers of the targeted communities |
| | Model | In many ways, this narrative is the culmination of some of the other narratives, in that it acknowledges the fear-based role of Narratives 3 and 4 (and seeks to provide farmers with the means to deal with a '*Fear of Change*', '*Fear of Peer Group Judgement*', and '*Fear of Being Different to Others*'), while highlighting the risk proposition of a perceived high '*Cost of Conversion*' or the '*Lock-In of Farmers with High Debt Levels*', and the difficulties of nudging people towards new ways of doing things. The '*Lack of Examples/Evidence*' and '*Inertia of Existing Knowledge Structure*' also play a role here, and in part, this model of watershed community conversion seeks to address those variables to provide the incentives necessary for farmers to change. If this 'narrative' was true, then '*Catchment-Scale Community Building*' becomes the primary lever, and the model would need to reflect the various relationships that either exist or that can (reasonably) be built in order to drive change in the system. More, stronger arrows would need to emerge from this variable to influence (directly or indirectly) the most central variables of the system, including 'Institutional Reticence to Adopt and Educate', '*Valuing Natural Capital in Multiple Dimensions*', '*Cost of Converting to Regenerative*', '*Gov't Policy Discourages Regen Ag*', and '*Lock In' of Farmers with High Debt Levels*'. It could also bring to the table with the farmers a wide range of external parties with experience, knowledge, skills, finance, and other resources to address 'Lack of independent agronomy advice'. Such an effort might also provide the case study needed to address the '*Lack of Examples/Evidence*' and to realise the economies of scale needed to remove the financial limitations of '*Lack of Resources/Finance*', '*Resources for new regen practices*', and the '*Cost of Converting to Regenerative*'. |
| | Solutions | In recognition of different areas and regions requiring different land use and management approaches, this narrative recognises that solutions should be presented and tackled at the catchment and community scale. By having a community or catchment working together, an organisation can identify solutions that mutually benefit the organisation at multiple levels, including but not limited to increased production, better profit margins, stronger social ties, or greater environmental benefits. This could involve coordinating with local councils and farmers within the catchment and would necessitate an organising body to ensure the appropriate training, education, and support were being delivered for the community needs. By converting a whole community, one can benefit from the economies of scale that can be delivered as well as the combined expertise. Groups such as The Mulloon Institute are one example of what this solution and narrative could look like. |

## 5. Discussion

We divide our discussion into three sections. First, we reveal what our case study highlighted for those seeking to increase the adoption of RegenAg practices in Australia. Second, we explain what the experience of using FCM to study this issue means for the communication of complexity within PM practice, and we outline some of the advantages and limitations of remote facilitation methods and strategies that could be applied during and beyond the COVID-19 pandemic. Third, we discuss what can be learned and applied for future PM exercises seeking to improve the management of SES before concluding this section with the limitations of our study and what that means for future research opportunities.

### 5.1. Solutions for Regenerative Agriculture

This case study and the results of the workshop highlight that the actions needed to increase the adoption of RegenAg must break the current 'reinforcing' paradigm of conventional agriculture. Stakeholders, a mix of landholders, trainers, researchers, and advocates, drew on their experience and knowledge to identify relationships within the Australian agricultural paradigm. Currently, business, government, the market, and social pressures seem to spiral down together in a race to the bottom, with few existing relationships in the system to incentivise a transformation. Understanding these complex forces highlights the need for coordinated actions at the institutional, social, and individual levels, across immediate and long timescales (decades). It is vital that RegenAg advocates find the messages and actions that overcome any paralysis of action in individuals and in communities [82].

One notable point in the FCM is the lack of balancing arrows (red arrows) within the system without the presence of the 'levers'—the blue icons noting actions we can take. Many of these were discussed or identified during the Modelling stage of the workshop as possible 'solutions'. If the 'levers' or actions we can take to act in the system are removed, the number of balancing connections in our model reduces by nearly half, going from 48 to 27 negative relationships, meaning the ability of the system to deliver on the outcome of interest '*Improved Ecosystem Health*' becomes diminished. If there is a push for conventional agriculture, the system, as it is currently drawn, intensifies that push. This impetus can rapidly move in a 'race to the bottom', enshrining the dominant paradigm of conventional agriculture in a downward spiral as degrading land leads to more artificial inputs, leading to further degrading land and more money and incentives being put into the system to prop it up.

The supportive, even reinforcing nature of the agricultural paradigm and the relationships between entities (government, business, and consumers) has the potential to 'lock-in' these conventional agricultural practices, as it is difficult for RegenAg to break into those relationships. Without significant policies or actions to provide a balancing relationship, there is no obvious incentive to change. Climate change could be one incentive, as it presents a severe challenge to human society. However, its impacts are often unclear, disputed, or occur over the long term. Without making the severe consequences of conventional agriculture (through impacts on climate change, biodiversity, human health, or something else) immediately apparent, it is difficult for RegenAg to generate enough urgency to push through.

While it is important to note this may not necessarily reflect the reality of the system (those balancing feedbacks may or may not exist regardless of what is shown here), the fact that stakeholders in favor of RegenAg believe this to be true is striking. This insight leads to the question: how can those balancing relationships be introduced to the system? This question is of interest to financial institutions and governments, many of whom have already begun work in this area. To increase the adoption of RegenAg, institutions can begin to investigate these balancing feedbacks, including but not limited to examining how 'support from industry groups' can depress fear of judgment or how the 'vested interests of business' might be aligned with instead of against valuing natural capital. As an example of a balancing relationship, for example, determining where and how to value natural capital would be vital for those seeking to increase the adoption of RegenAg. Such efforts may also address the difficulties of "going green when you're in the red." These and other opportunities present a possibility for more balancing relationships within the system. The levers on the FCM also represent possible policy/intervention opportunities that workshop participants perceive as fundamentally relevant to a wider adoption of RegenAg in Australia. These opportunities are further discussed and explored in the five narratives. Notably, the solutions outlined under the five narratives coincide in the need for well coordinated, multi-scale (state; catchment, community) and multi-actor (federal, state, local government; industry; farmers and local communities) efforts to promote the desired shift from traditional to RegenAg practices. This need for work at various scales is

documented within the research and the RegenAg movement (Chapman, 2019; Gordon, 2020; Murphy, 2021). As noted by [21], there are a number of 'spheres' or scales in which to push for RegenAg, including the personal, the practical, and the political. Our work found a similar pattern. What is clear from our exercise is that among RegenAg practitioners, the role institutions play seems to matter to them a great deal, but they also note the interactions with social groups and personal identities and habits.

In addition, a number of barriers identified during the workshop, including an upfront cost to convert to regenerative, debt levels, lack of resources, and an ingrained view that environmental and economic outcomes cannot both be achieved, all suggest that a transition to RegenAg is expensive. This expense may or may not reflect reality, particularly when considering the return on investment and resilience offered by many RegenAg practices, but the 'perception' of the expense seems to be important for those considering a transition.

The FCM process also noted behavioral traps and pain points of individual farmers, which are perhaps just as difficult, if not more so, to change than government policy. This includes *'Fear of change'*, *'Fear of Being Different...,'* and *'Fear of judgment from peers'*. As an area perhaps less explored within agricultural policy, but with a growing body of research on the importance of stakeholder outreach and 'tailored' communication [83,84], there is a potential to change 'faster' through education and outreach. We, like others [21], advocate that education and outreach should centre on the personal sphere, aiming for critical awareness [85], reflection [86,87] and transformative learning [88,89] that allows for the deeper questioning and altering of underlying values and beliefs [68].

Based on the FCM that was elicited, the narratives derived from it, and follow-up discussions with workshop participants, we identify several recommended areas of focus to improve the adoption of RegenAg in Australia at various scales:

**Narrative 1: Government First**—If 'Big Government' is the problem, then 'Big Government' must be a part of the solution. This includes a coordinated effort from federal and state governments. While noting the effect the individual voters and media have on the government, if this narrative is true, a drastic reform of government policy is needed. More incentives need to be provided for a switch to RegenAg, and they could show up in a reform of drought policy or an integration of water and soil acts, which are perhaps more in line with watershed boundaries as opposed to arbitrary geopolitical ones. Stakeholders noted that not only does this commitment need to be significant, it also needs to be 'long-term' in order to align with the cycles of natural capital and to "give confidence to land managers, industry, educational institutions, NGOs, and the broader public." Part of that effort could then include greater efforts to inform the voting public of RegenAg interests and actively push Parliament to embrace policies benefiting RegenAg by direct lobbying from RegenAg advocates and practitioners towards the relevant departments and ministers.

**Narrative 2: The Market Matters**—Find ways to increase consumer demand for products of RegenAg (affecting 'Consumer Demand for Cheap Food' and 'Market Demand' for "Clean and Green"), which could happen in a number of ways:

- Provide government incentives to subsidise the cost of regenerative products, either in out-of-the-gate packaging and production, or in reducing the high upfront costs needed to switch to regenerative.
- Create regional processing and distribution centres in high agricultural areas devoted to regenerative products and lowering costs by producing at scale.
- Incentivise supermarkets to carry regenerative products either at lower prices or in high-value locations in stores to encourage more sales.
- Increase funding to marketing and advertising to craft a more compelling narrative for regenerative products to direct-sell to consumers

**Narrative 3: Pressured Communities**—'Normalise uptake' of RegenAg practices to remove any social stigma that comes from such a transition or practice. Being able to point to socially accepted indicators of success, or cultural capital [18,80,81] for regenerative farmers, such as increased income or production, can help shield such farmers from criticism. Therefore, building up evidence and case studies to complement these transitions,

ideally over the long term, can help. In addition, identifying and working with local and community champions, which could include members of local councils, NGOs, or fellow farmers, could present additional social visibility and support. Providing training and support and building "communities of practitioners and networks of conversation" can also assist and can span across regions due to the access and ease of the internet and social media channels.

**Narrative 4: Start with People**—Appropriate solutions would need to address the fear that underlies much of the social and institutional resistance. 'Inducing epiphanies' as sought by [21] would be crucial. Education and outreach to converse and and engage in dialogue with skeptics would be central to this effort, with the ultimate aims of transforming the hearts and minds of the agricultural industry, highlighting the "hope, dreams, and aspirations" of "leaving the land in better shape for the next generation". This would also include better marketing and targeting of consumers to have them switch to products of RegenAg. However, the timescale for this scale of change, as noted by participants, is likely to take decades.

**Narrative 5: Community by Community**—In recognition of different geographical areas and bio-regions requiring different land use and management approaches, this narrative recognises that solutions should be implemented both at the catchment and community scale. By having a community or catchment working together, an organisation (non-profit or even a government department) can identify solutions that mutually benefit the organisation at multiple levels, including but not limited to increased production, better profit margins, stronger social ties, or greater environmental benefits. This could involve coordinating with local councils and farmers within the catchment and would necessitate an organising body (such as TMI) to ensure that the appropriate training, education, and support were being delivered for the community needs. By converting a whole community, one can benefit from the economies of scale that can be delivered as well as the combined expertise.

The bottom line is that it is critical to engage stakeholders in the adoption of RegenAg, as it is both a context-specific area of practice within the limits of the land and, crucially, the adoption of RegenAg is a personal and social issue [23,90,91]. We believe therefore that building on the current study, perhaps by further investigating the validity of the narratives and their implications, could identify further actions to take to improve adoption as well as highlight additional barriers that the movement may face and that were not apparent to the participants of our FCM workshop. Blindspots in the FCM could be illuminated by input from and conversations with the voices and perceptions from conventional agriculture. By understanding the focus of different (and at times opposing) stakeholder groups, PM practitioners could focus future workshop discussions on those actions and policies upon which there is both broad consensus and a sufficient evidence base to operate. FCM was a suitable tool for us to use to negotiate this effort in a small sample size, and it may be worth exploring in subsequent PM exercises with other stakeholders of Australian agriculture, including conventional farmers.

A note of caution is due here, since this was one case study with a limited number of participants, and the participants were largely in favour of RegenAg. It is important to bear in mind the possible bias in these responses. With a small sample size, the findings might not be representative of the system at large, and these results therefore need to be interpreted with caution. However, the PM exercise that is reported here points to narratives of interest and identifies several variables and relationships worthy of further investigation. Each of these narratives could be improved and/or challenged by asking questions such as "How do we know this to be true?", "What would need to be seen for it to be proven true/false?", "What might some indications be to show that we are wrong?" However, these critical questions cannot be asked if there is nothing to explore in the first place.

Another source of uncertainty is the inherent complexity of the socio-ecological system we are studying here. A farm is subject to consumer demand, market prices, government

policy, social pressures from peers and family, environmental disasters, long-term climatic trends, access to education and research, and the struggle to get up early in the morning. It is impossible to know with absolute certainty the status of some of these variables or the nature of their relationships. In this sense, the model here does not, and cannot, perfectly reflect reality (the map is not the territory, [92]). Not only that, but some of these barriers, particularly institutional ones, are not only complex but slow moving, requiring huge efforts and investments to drive change over the long term. Left untouched, this seemingly insurmountable challenge could be discouraging. Complexity and uncertainty, however, cannot be an excuse for inaction. The identification of key variables and relationships, as we have done here, provides one path forward to asking better questions and finding more targeted actions. The PM exercise and the FCM produced provides a blueprint of the first steps we should be taking to untangle the complexity and uncertainty of the system in attunement with people's beliefs and perceptions of how the agricultural paradigm operates. With these new insights in hand, our knowledge of the system becomes a little more complete, and we can work with stakeholders to look for new leverage points for change.

### 5.2. Reporting FCM

Our FCM analysis was useful for understanding the various barriers that at present seem to limit the increasing adoption of RegenAg practices in Australia as well as identifying the opportunities that exist for RegenAg going forward. Unsurprisingly, our FCM highlights the complexity of this issue, the various scales at which change needs to happen, and the multitude of relationships and actors that need to be involved. Our analysis showed that our stakeholder group, as a pro-RegenAg group, is aware of these complexities, and we were able to work together to demonstrate this understanding. While we did not include any 'voices' from conventional agriculture, this understanding is a useful entry point for those promoting more sustainable practices in Australian agriculture and offers a platform for outreach to different, even 'opposing', voices.

From the analysis of the FCM, we are able to make suggestions about variables and relationships of interest. Our centrality analysis highlighted several key variables, and these are '*Institutional Reticence to Adopt and Educate*', '*Valuing Natural Capital in Multiple Dimensions*', '*Cost of Converting to Regenerative*', '*Gov't Policy Discourages Regen Ag*', and '*Lock In' of Farmers with High Debt Levels*''. However, for us, the promise of FCM was in its ability to serve as a 'boundary object' around which our discussion of RegenAg adoption could be mediated, negotiated, and navigated. As we built our model, using MURAL and Zoom, the 'visual' nature of the process helped us to represent the connections between abstract concepts, any of which could be changed by proposals from any stakeholder [93]. Used in this way, our FCM, in keeping with the literature, helped stakeholders to: (1) create a shared language, allowing for diverse knowledge and different disciplines to work together by creating a 'model' together on MURAL and in iterations afterwards; (2) clearly identify points of difference or similarity of relationships within and between stakeholder mental models; and (3) "transform their current collective knowledge toward an agreement of facts through discussion, negotiation, and careful scrutiny of what they know" to move towards innovation, cooperation, and consensus building [94–96]. PM and the process we used provided a model by which we could encourage individuals to share their perspectives on a collective issue of interest, forming an explicit group mental model as the object around which the discussion could be mediated [75,97,98]. Even if stakeholders did not completely agree with the model, it is more difficult to ignore the model "because they built the model themselves" [43,99].

We complemented this use of the model as a boundary object with the development of 'narratives' to report the findings of our FCM in a way that facilitates the communication of complexity of this issue and the system(s) it involves. While each 'narrative' is a simplification of reality, the presence of multiple stories actually allows us to embrace complexity and to communicate it [63,64,100,101]. By showing a number of possible,

plausible narratives, we demonstrate that there is more than one way to see and interpret the system, helping participants to acknowledge that their interpretation is not the only one. Even if valid, their chosen narrative is probably imperfect or incomplete. As such, we consider narratives as an instrument to communicate complexity that is constantly evolving and under construction, aligning with the iteration so desired and so necessary in an overall PM process. In short, FCM creates a visualisation, and together with the accompanying reporting and communication of the results as narratives, this process of model building with stakeholders makes the complexity of this system starkly apparent. Recognition of that complexity is the first step to finding the leverage points needed to transform that system [68].

*5.3. Limitations*

While the findings of our study build on work in RegenAg and contribute to PM in several ways, there were also limitations. Being limited to one case study and one workshop meant that a cross-case analysis was not possible. No PM, even with the same stakeholders, is the same, and this makes the presence of a 'control' group impossible. However, a workshop similar in intent with different stakeholders could have been interesting. In a similar fashion, a workshop focused on a different topic (for example, groundwater usage) could have provided additional insights and comparisons between how different workshops were both run and received and what insights were produced as a result. For example, for this workshop, we introduced two possible metrics before the workshop. While neither was ultimately chosen as the metric of the system, this may still have introduced bias by framing the conversation.

Additionally, this study was intentionally absent of conventional agriculture producers. While a purposeful choice to minimise disruptive and destructive conflict, those 'voices' were not considered in our depiction of the 'barriers' of RegenAg adoption. Indeed, the workshop was based on the assumption that more RegenAg is desirable, to which not all farmers might agree. However, the presence of a largely homogenous, pro-RegenAg group did allow for us to notice some of the in-group distinctions and conflicting perceptions. We stand by the decision to proceed as we did, but transparency dictates noting this absence, and future work can and should find ways to include more 'conventional' participants, which could add more representativeness and complexity to our FCM. Another consideration for our group was who was left out by the move to virtual. It was a necessary move under conditions of social distancing imposed by the pandemic, but it did exclude those without internet access or the skill, and it may have even limited full participation from our participants who may have struggled to navigate the new technology. These limits should be considered and accounted for in future studies of virtual PM.

Our small sample size also did not allow for a 'full' representation of RegenAg, as it is not a homogenous group. RegenAg is an umbrella, covering a variety of practices and outcomes [27], and working with a different group of participants may have led to a different model. The small sample size also limits the degree to which statistical analysis can find 'significance' in the strength of relationships on our model. Future work could combat this by creating individual maps with participants before aggregating into a larger 'group' map or a separate larger group discussion. We could not take this approach with our limited time and resources, but we hope to do so in the future. In addition, sample size is a pervasive issue in PM, but it is one that has the potential to be overcome with virtual facilitation (even beyond COVID-19), as shown here. Under normal circumstances, COVID-19 and social distancing restrictions would have made this engagement impossible, but we managed to pivot to a successful virtual delivery under lockdown, and that same method could be used to expand sample sizes as geographical distance or the physical size of the room are no longer limiting factors.

### 5.4. Future Research

Notwithstanding the above limitations, our case study suggests important areas of future development and research. The first is that for those seeking to increase RegenAg, we should further investigate the narratives presented along with their accompanying solutions to ascertain if they are 'true' in the eyes of stakeholders and in reality, as 'truth' in one does not guarantee 'truth' in the other. Our aim is for narratives to be the best representation of reality we can have at a certain point in time given the information available. A good narrative should be accepted by as many stakeholders as possible, meaning that the narrative speaks to multiple perspectives, and possess multiple, legitimate sources of corroborating evidence. This entails examining the relevant variables and their relationships to ascertain if available evidence supports their existence. For example, for **Narrative 1: 'Government First'**, to determine to what extent "*Government policy discourages RegenAg*", a policy 'wish list' of RegenAg stakeholders could be created during a follow-up a PM workshop, online or offline iteration (Google Docs, email, etc), or by compiling and synthesising reports from various interest groups—this could be compared to current policy to identify where the gaps are and what is needed to address them. Bespoke follow-up activities could be implemented for each of the narratives, involving stakeholders at various stages to further advance and solidify the opportunities identified during the initial engagement.

Second, our study suggests that the use of FCM (or any PM technique) in a virtual format should be explored further, particularly considering the current COVID-19 contingency. Stakeholder engagement during the COVID crisis offers a unique opportunity to benchmark remote facilitation designs against face-to-face workshops, with the aim of providing general recommendations on when either method would be preferable (e.g., depending on problem situation, level of stakeholder conflict, types of SES resources involved, etc.). However, it is also worth noting here who may be excluded by such a move to virtual, including those without the internet access or technological skills needed to successfully participate in this online format. Based on the experience of the activities conducted as part of this case study, a virtual approach has the potential to offer benefits in coordinating participatory processes of large stakeholder groups spread over vast geographical areas, which may be useful in many other SES issues.

Previous studies using FCM often begin with individual maps before aggregating to a larger collective map, usually over the course of several meetings [43,47]. We did not draw individual models with stakeholders one-on-one, instead waiting to develop a full group map in one sitting due to our time constraints imposed by the pandemic, with individual follow-up as needed. While not inherently a negative approach, it is interesting to note the difference and the possibility of further depth and complexity of the FCM presented by iterations between individual maps and collective maps, depending on where the starting point may be. The use of our 'narrative' templates (story, actors, model, solutions) may be an alternative way to guide this process, collecting stories beforehand (which can be accompanied by individual cognitive maps) and then iterating these narratives side by side with the main FCM over the course of PM. Logic Models—a framework for charting the links between a project's resources, activities, and outputs and its intended outcomes—could also be used to better structure and sequence the activities needed to build FCMs.

Finally, to develop a full picture of the dynamics people bring into any participatory or PM workshop, we should further explore the idea of what would constitute an ideal 'enabling' environment of PM (i.e., specify the articulation of tools, facilitation techniques, communication strategies, and follow-up activities that should be implemented to deliver on the promises of PM) and its ability to construct and manage PM as a socio-emotional system from which to elicit and transform individual mental models. This aspect of PM is often missing from reporting on PM, which tends to focus on the model. The dynamics of the emotions and social structures that stakeholders bring with them into a workshop holds implications for the way we conduct PM, the way we report the findings, and the way

we seek to address issues of SES (such as increasing the adoption of RegenAg practices). 'Narratives' seem to be one of many possible tools that can assist in the PM process, as it can effectively align the insights developed from a scientific modelling exercise with the ways people best assimilate information. Similar to narratives, there are likely to be other tools out there, and we should look to further explore how concepts, ideas, and practices from various behavioral sciences can improve any aspect of the design, practice, or evaluation of PM—so that it is behaviorally attuned to the problems that it is trying to solve.

## 6. Conclusions

Taken together, the findings of this paper have a number of important implications both for RegenAg in Australia and the future practice of PM. In our case study, the co-construction of the FCM revealed the myriad of connections at play in the adoption of RegenAg in Australia, highlighting the complex forces at work and the need for coordinated actions at the institutional, social, and individual levels, across long timescales (decades). Such actions are necessary for RegenAg to play a greater role in Australia's agricultural paradigm and bring 'balancing' relationships to a system currently reliant on conventional agriculture with few internal incentives to change. The crisis of climate change and a degrading environment may be the ultimate reasons for change, but in addition to communicating the severe danger of these crises, RegenAg advocates must also find the messages and actions that overcome any paralysis of action in individuals and in communities [82].

We also used 'narratives' to identify some of these solutions, communicate with our stakeholders, and report some of our findings [63]. The simplicity of narratives (as a product derived from a PM workshop) makes it a replicable and repeatable format that should be used in the future for the practice of PM. Additionally, PM practice may also be improved by the use of our 'virtual' facilitation approach, as we were able to successfully build an FCM, despite participants being hundreds of kilometers apart from each other and from the research team.

By using 'narratives' to communicate the results of our FCM, we can tailor our findings to a format well-suited for communicating complexity. This format is compatible with the existence of various (and sometimes competing) narratives—more than one may be present, and the priority for any particular narrative may shift. This is where tools such as FCM and narratives can work together. The narratives inform the FCM in looking at what variables and relationships to explore further, and the FCM informs the narratives by showing what is possible and what those relationships might mean for actions in the real world. Although our study suggests that this combination of storytelling and PM has enormous potential, more work can and should be done here, particularly in exploring how narratives might be developed into a consistent, repeatable framework to work alongside the PM process from start to finish. In this paper, we argue that understanding individual farmer perceptions should precede and accompany any effort to close the gap between barriers and opportunities for the adoption of RegenAg practices, or indeed, any other practice that improves the state of a SES. This means that any effort to increase the adoption of such practices should first consider how individuals make day-to-day decisions by determining which narratives are dominant. Developing awareness and attunement to these narratives has helped identify those factors that undermine or shape potential solutions and how limiting or deep-seated beliefs might be countered by articulating different problem definitions and framing different solution options. In short, instead of starting with models and policy, PM practice should start with people based on modern understandings and science of what they think and behave.

**Author Contributions:** Conceptualization, D.C.K. and J.C.-R.; methodology, D.C.K. and J.C.-R.; formal analysis, D.C.K. and J.C.-R.; writing—original draft preparation, D.K; writing—review and editing, D.C.K. and J.C.-R.; supervision, J.C.-R.; All authors have read and agreed to the published version of the manuscript.

**Funding:** This research was funded by The Australian Research Council (ARC) Discovery Project grant number DP190101584.

**Institutional Review Board Statement:** Not applicable.

**Informed Consent Statement:** Informed consent was obtained from all subjects involved in the study.

**Data Availability Statement:** Not applicable.

**Acknowledgments:** We acknowledge The Mulloon Institute who provided the institutional support and testing of participatory modelling methods presented in this paper.

**Conflicts of Interest:** The authors declare no conflict of interest.

## Abbreviations

The following abbreviations are used in this manuscript:

| | |
|---|---|
| PM | Participatory Modelling |
| SES | Socio-Environmental Systems |
| FCM | Fuzzy Cognitive Mapping |
| TMI | The Mulloon Institute |
| NSF | Natural Sequence Farming |

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
