# Peer review of "What Prevents the Adoption of Regenerative Agriculture and What Can We Do about It? Lessons and Narratives from a Participatory Modelling Exercise in Australia"

_land, doi:10.3390/land11091383_

Round 1

Reviewer 1 Report

The article “What prevents the adoption of regenerative agriculture and what can we do about it? Lessons and narratives from a participatory modelling exercise in Australia” presents and discusses the results of a Participatory Modelling exercise with Regenerative Agriculture stakeholders in Australia. The aim was to provide a blueprint of how challenges and opportunities could be collaboratively explored in alignment with landholders personal views and perspectives. Authors provide a comprehensive description of Fuzzy Cognitive Maps (FCM) methodology and the way they follow to unpack and formalise landholder perspectives into a semi-quantitative shared mental model of the barriers and enablers for adoption of Regenerative agricultural practices, and to subsequently identify actions that might close the gap between the two. They also constructed five dominant narratives which encode the key drivers and pain points in the system identified and extracted from the FCM to promote the internalisation of outcomes and lessons from the engagement.

As it is now, the paper is well organized and developed. Both methodology and results and discussion are well structured and provide full details to the readers. To be noted the transparency in highlighting the shortcomings. No one is left behind.

Few comments:

PM and SES acronyms should be first introduced prior to be used.

Line 177: a reference has a question mark

Line 215: there is a typo (to/to).

Good luck!

Author Response

To Our Editor and Reviewers,

Thank you so much for your comments. This paper has been greatly improved by your feedback. We have tracked our changes below, focusing on the issues you have alluded to, shortening the paper by ~10% by removing (old) section 5.3 which contained a number of redundancies which are discussed elsewhere in the paper. We thank you for taking the time to read and review our paper. 

Warm Regards,

Danny and Juan 

Reviews

R1 Comments

The article “What prevents the adoption of regenerative agriculture and what can we do about it? Lessons and narratives from a participatory modelling exercise in Australia” presents and discusses the results of a Participatory Modelling exercise with Regenerative Agriculture stakeholders in Australia. The aim was to provide a blueprint of how challenges and opportunities could be collaboratively explored in alignment with landholders’ personal views and perspectives. Authors provide a comprehensive description of Fuzzy Cognitive Maps (FCM) methodology and the way they follow to unpack and formalise landholder perspectives into a semi-quantitative shared ‘mental model’ of the barriers and enablers for adoption of Regenerative agricultural practices, and to subsequently identify actions that might close the gap between the two. They also constructed five dominant narratives which encode the key drivers and pain points in the system identified and extracted from the FCM to promote the internalisation of outcomes and lessons from the engagement.

As it is now, the paper is well organised and developed. Both methodology and results and discussion are well structured and provide full details to the readers. To be noted the transparency in highlighting the shortcomings. No one is left behind.

Author’s response:  

We thank R1 for the positive assessment of our paper

Few comments:

R1.1 —PM and SES acronyms should be first introduced prior to being used.

Author’s response: 

Thanks for pointing this out. We have defined both acronyms on the main text prior to them being used. Also, to keep track of the various acronyms used in this paper, we have added a glossary of terms at the end of the manuscript (Abbreviations section) as suggested in R3.3 below. 

R1.2 —Line 177: a reference has a question mark

Author’s response: 

Fixed

R1.3 —Line 215: there is a typo (to/to).

Author’s response: 

Fixed

R2 Comments

R2.1 —The author should supplement the full name of "TMI" in Line 134 of the main body of the manuscript (Page3, Line 134). Abbreviations used in the article should be noted at the first appearance, not just in the legend.

Author’s response: 

This issue was also identified by R1 and R3. Agreed and fixed as per our response to R.1.1 above. 

R2.2 — Fig 1 on the right side of the picture corresponds to "Activities include building leaky weirs, contour Banks, embankments, and vegetation plantings"? If so, different activity pictures should be annotated to help readers better understand the content of different activities (Figure 1 legends).

Author’s response: 

Thanks for pointing this out. We have followed the suggestion and added annotations to the four picture panels in Figure 1.

R2.3 — The author should add the meaning of "SES" (Page 5, Line 165). Abbreviations used in the article should be explained on the first appearance.

Author’s response: 

This issue was also identified by R1 and R3. Agreed and fixed as per our response to R.1.1 above. 

R2.4 — Will everyone in the experiment develop their own "FCM"? Is the weight of "FCM" formed by different groups the same? How do I define the group in this article? (Page 5, Lines 183-185)

Author’s response:

Details of how the experiment was set up are provided in section 3.2. In short, participants were randomly split into two groups. Each group developed their own FCM, which were “woven” together into one model after comparing to find common variables and aggregate them, re-visiting the recordings and transcripts of the workshop. This is explained in section 3.2.5.

R2.5 — Maybe I didn't find it, please explain "the most relevant 'variables'" (Page12, Line451).

Author’s response:

This simply refers to a standard node importance analysis using network centrality metrics. We have clarified this in the text, pointing to section 4.1.2 which contains the details of this analysis.

R2.6 — May I ask what magnitude of data the author needs to collect to make the model reliable in the process of building the model?

Author’s response:

There is no straightforward answer to this question. In Participatory Systems Mapping exercises we usually go through several rounds (usually 2 or 3) of engagement with the stakeholders, progressively and iteratively completing and validating the FCM with the aim of converging to a “shared” mental model of the system. In this study we developed two FCMs which were combined (woven) into a single model after being verified and validated by stakeholders (post-workshop). 

The more iterations, the more reliable the model becomes, but this comes at a cost e.g., stakeholder engagement fatigue, diminishing returns on what is being added to the model, etc. For example, a workshop similar in intent with stakeholders opposing/unconvinced with RegenAg could have been interesting, providing alternative views to incorporate into the FCM. Unfortunately, as is usually the case in these types of engagements, opportunities to iterate are few and far between (due to funding, stakeholder availability, time constraints, etc.). Ideally, the model should be revisited and updated on an ongoing basis, to make sure it reflects the current state of the system. 

We have added some clarifying remarks in section 5.3 (Limitations)

R3 Comments

R3.1 — line 99 - dispersed, not "disperse".  SES acronym should be defined.

Author’s response: 

Fixed. The acronym issue was also identified by R1 and R2. Agreed and fixed as per our response to R.1.1 above. 

R3.2 — line 121 - "partnering with microbes, etc." - I understand the sentiment, but is this the right word to describe the relationship?

Author’s response:

Our preference would be to keep the term “partnerships” as it is a powerful way to describe the nature of the relationships that are being pursued by Natural Sequence Farming.

R3.3 — line 134 - Project and TMI --- need to explain what these are.  The Mulloon Initiative?  What is the Project?  Perhaps you need a brief glossary of terms since there are a lot of acronyms to keep track of in this paper.

Author’s response:

The acronym issue was also identified by R1 and R2. Agreed and fixed as per our response to R.1.1 above.

R3.4 — line 173 - built upon by whom?

Author’s response: 

Thanks, the author’s surname was accidentally omitted and has been added now.

R3.5 — At around line 292, it occurred to me that your process might have been improved by including some concepts from "logic models".  Logic models are useful in structuring somewhat linear processes and clearly distinguish between outputs and outcomes.  Perhaps this might be mentioned in your conclusions and recommendations.  

Author’s response:

Thanks for pointing this out. Logic Models were not in our radar as they are not commonly associated with Participatory Modelling tools. Logic Models seem to be useful for planning purposes (commonly used to design education programs), offering a framework for charting the links between a project’s resources, activities, and outputs and its intended outcomes. Although the application of Logic Modelling concepts to Participatory Modelling is beyond the scope of this paper, we believe it could provide a simple and clear template to structure and sequence activities leading up to the construction of an FCM. Such a template could be very useful when facilitating a Participatory Modelling workshop remotely using video conferencing software and digital whiteboards. 

We have added a brief mention to the above in the section 5.4 (Future Research)   

R3.6 — lines 377-379 - confusing sentence.  Likely needs to be rewritten.

Author’s response:

We rephrased this sentence for clarity.

R3.7 — lines 383-389 - starts with "in short"; but, this sentence is way too long.  

Author’s response:

We shortened and rephrased this sentence for clarity.

R3.8 — line 405 - you provide a citation, but should likely list the author in the text itself.  

Author’s response:

Agreed and added.

R3.9 — It's apparent that the sections I just read (debrief and followup) are written primarily by a different author.  Readability and flow would be improved if the lead author on the first sections would review and suggest changes to these sections.  

Author’s response:

Sections 3.2.4 and 3.2.5 were written and revised in the same way as the previous sections, and we cannot identify substantial changes in the tone or narrative. In any case, we have completely revised the two sections for clarity and flow.

R3.10 — line 412:  Again, I think the author ought to be listed.  But, this is an editorial decision.

Author’s response:

Agreed and added.

Editor Comments

E1 — I have concerns that it might be a bit too long and dense for the common reader. There are likely quite a few redundancies and complexities that could be streamlined, reducing the length. I'd suggest that the editor challenge the authors to reduce it by 10% in an attempt to accomplish this.

Author’s response:

We thank the Editor for pointing this out. Upon reviewing the manuscript we have decided to remove (old) section 5.3 (PM practice in value-laden SES) which indeed had a number of redundancies and complexities that are mentioned elsewhere in the paper. This has not only reduced the length of the paper by ~10% but also establishes a more direct flow between the outcomes of the participatory modelling process, the discussion on limitations, and future research.

Reviewer 2 Report

1. The author should supplement the full name of "TMI" in Line 134 of the main body of the manuscript (Page3, Line 134). Abbreviations used in the article should be noted at the first appearance, not just in the legend.

2. Fig 1 on the right side of the picture is corresponding to "Activities include building leaky weirs, contour Banks, embankments, and vegetation plantings"? If so, different activity pictures should be annotated to help readers better understand the content of different activities (Figure 1 legends).

3. The author should add the meaning of "SES" (Page 5, Line 165). Abbreviations used in the article should be explained on the first appearance.

4. Will everyone in the experiment develop their own "FCM"? Is the weight of "FCM" formed by different groups the same? How do I define the group in this article? (Page 5, Lines 183-185)

5. Maybe I didn't find it, please explain "the most relevant 'variables'" (Page12, Line451).

6. May I ask what magnitude of data the author needs to collect to make the model reliable in the process of building the model?

Author Response

(The authors gave the same response as above.)

Reviewer 3 Report

line 99 - dispersed, not "disperse".  SES acronym should be defined.

line 121 - "partnering with microbes, etc." - I understand the sentiment, but is this the right word to describe the relationship?

line 134 - Project and TMI --- need to explain what these are.  The Mulloon Initiative?  What is the Project?  Perhaps you need a brief glossary of terms since there are a lot of acronyms to keep track of in this paper.

line 173 - built upon by whom?

At around line 292, it occurred to me that your process might have been improved by including some concepts from "logic models".  Logic models are useful in structuring somewhat linear processes and clearly distinguish between outputs and outcomes.  Perhaps this might be mentioned in your conclusions and recommendations.  

lines 377-379 - confusing sentence.  Likely needs to be rewritten.

lines 383-389 - starts with "in short"; but, this sentence is way too long.  

line 405 - you provide a citation, but should likely list the author in the text itself.  

It's apparent that the sections I just read (debrief and followup) are written primarily by a different author.  Readability and flow would be improved if the lead author on the first sections would review and suggest changes to these sections.  

line 412:  Again, I think the author ought to be listed.  But, this is an editorial decision.

Author Response

(The authors gave the same response as above.)
